# The Burden of Gastric Cancer Attributable to High Sodium Intake: A Longitudinal Study from 1990 to 2019 in China

**DOI:** 10.3390/nu15245088

**Published:** 2023-12-13

**Authors:** Liying Jiang, Anqi Wang, Shuo Yang, Haiqin Fang, Qihe Wang, Huzhong Li, Sana Liu, Aidong Liu

**Affiliations:** 1Jiading Central Hospital, Shanghai University of Medicine & Health Sciences, Shanghai 201899, China; j_meili@126.com; 2Department of Prevention Medicine, College of Public Health, Shanghai University of Medicine & Health Sciences, Shanghai 201318, China; 3Graduate School, Shanghai University of Traditional & Chinese Medicine, Shanghai 201203, China; wanganqi202309@163.com; 4Department of Health Inspection and Quarantine, College of Public health, Shanghai University of Medicine & Health Sciences, Shanghai 201318, China; 13124259192@163.com; 5Department of Nutrition Division I, China National Center for Food Safety Risk Assessment, Beijing 100022, China; fanghaiqin@cfsa.net.cn (H.F.); wangqihe@cfsa.net.cn (Q.W.); lihuzhong@cfsa.net.cn (H.L.); liusana@cfsa.net.cn (S.L.); 6National Institute for Nutrition and Health, Chinese Center for Disease Control and Prevention, Beijing 100050, China

**Keywords:** gastric cancer, disease burden, high sodium intake, China

## Abstract

Backgrounds: Excessive intake of sodium is a crucial risk factor of gastric cancer. However, it is still unclear whether the profile of gastric cancer burden is attributable to high sodium intake in China. This study aims to evaluate the levels and trends of gastric cancer burden attributable to high sodium intake across China from 1990 to 2019. Methods: We acquired data from the GBD (Global Burden of Disease Study) 2019 via the Global Health Data Exchange query tool. The details of regions from 1 January 1990 to 31 December 2019 from the China National Center for Food Safety Risk Assessment were also used. We conducted an integrated analysis on the gastric cancer burden attributable to high sodium intake among Chinese residents. The gastric cancer-related deaths, disability-adjusted life years (DALYs), age-standardized mortality rate (ASMR), and age-standardized DALYs rate (ASDR), all being calculated to be attributable to sodium intake, were reckoned as separated by age, sex, SDI, and regions. Then, the estimated annual percentage change (EAPC) was regarded as the secular trends of gastric cancer’s ASMR and ASDR due to high sodium intake from 1990 to 2019. We further explored the associations between SDI (Socio-demographic index) and the ASMR and ASDR. The rates were calculated per 100,000 population as age-standardized rates. Results: Briefly, the number of gastric cancer-related deaths and DALYs being attributed to high sodium intake were 37,131.48 (95% UI: 833.14 to 138,478.72) and 873,813.19 (95% UI: 19,283.13 to 3,220,231.82) in 2019; both have increased by a third since 1990. However, the ASMR decreased with an EAPC of −1.72% (95% CI: −2.11% to −1.33%), while ASDR increased with an EAPC of 0.36% (95% CI: 0.08% to 0.68%), respectively. The age-specific numbers and rates of deaths, as well as DALYs of gastric cancer being attributed to high sodium intake, elevated gradually with age. And, they were higher in males than in females. The gastric cancer burden being attributed to high sodium intake in 2019 and its temporal trends from 1990 to 2019 varied greatly by SDI quintile and geographic locations. There was a strong negative association between the EAPC in ASMR and SDI in 2019 (ρ = −0.642, *p* < 0.001). The EAPC in ASDR and SDI also exhibited a negative connection in 2019 (ρ = −0.538, *p* = 0.0012). Conclusions: Overall, using a longitudinal sample from different regions, the study presented that gastric cancer burden attributed to high sodium intake still exists seriously and varies remarkably by regions, sex, and age across China. The disparity of socioeconomic status on disease burden also exists. Integrated and precise approaches for gastric cancer prevention are anticipated in the future.

## 1. Introduction

As a common malignant tumor, gastric cancer has sparked concern and posed a serious disease burden [1]. It is a leading cause of death and the fourth-highest associated with cancer. It is the fifth most diagnosed cancer worldwide, accounting for 1.1 million new cases (representing 5.60% cancer cases) and 7.6 hundred thousand new deaths (representing 7.70% cancer cases) in 2020 [2]. According to the 2018 global cancer statistics, 456,124 new gastric cancer cases and 390,182 deaths were speculated to have amassed in China, accounting for 49.9% of the gastric cancer patients in the world. The serious situation highlights the necessity and urgency of the prevention and treatment of gastric cancer in China [3].

Compelling studies have confirmed that gastric cancer is a complex disease caused by infectious factors (*H. pylori* infection), environmental risk factors, and lifestyles. Specifically, dietary factors were associated with an increased disease risk, such as specific food propensity (meat and alcohol), an unhealthy diet with too much salt, a low vitamin A and C diet, and a large amount of processed food and smoked food [4,5]. A meta-analysis of 268,718 subjects showed that high or moderate salt intake had a 0.68-fold and 0.41-fold increased risk of gastric cancer compared with a group with low salt intake [6]. The popularity and detrimental effect of a high sodium diet accelerate the development of gastric cancer. The results of the Global Burden of Disease Study (GBD) show that the disease burden attributed to a high-sodium diet accounts for more than 40% of the burden and is associated with poor dietary sodium intake [7]. Although the worldwide gastric cancer burden being attributed to high sodium intake has been previously recognized based on the data from the Global Burden of Disease Study in 2019, the temporal profiles and the spatial patterns of the disease burden being attributed to high sodium intake may differ significantly from the patterns in China [7].

According to research on dietary sodium sources of Chinese residents, dietary sodium mainly comes from added salt in family cooking, followed by soy sauce and processed foods [8]. In 2012, results of the nutrition and health status of Chinese residents showed that the dietary sodium intake was as high as 5013 mg/d, far exceeding the recommended amount of the world health organization (WHO) (2000 mg/d); it was also higher than the global average level of dietary sodium intake (3950 mg/d) [9]. Over 80% of the residents’ dietary sodium intake exceeded the recommended level [10].

High sodium intake is linked to lots of gastric cancer cases and also contributes to the disease burden. Previous evidence suggests that excessive salt consumption that could promote *Helicobacter pylori* (*H. pylori*) colonization in the area of stomach is a well-known risk factor for the development of gastric cancer [11]. H. pylori infection constitutes an identified risk factor of the disease, with substantial impact on populations with a habit of high sodium intake in daily life. A comprehensive mechanism brings its superiority into full play by cultivating gastric adenocarcinoma development, owing to the synergy between high sodium intake and *H. pylori* colonization. Collectively, the multitude of mechanisms encompass the disruption of mucosal barriers, cellular integrity, *H. pylori* gene expression modification, oxidative stress induction, and inflammatory responses provocation. The gastric cancer burden caused by high sodium intake poses a great challenge to public health and has continued to increase in recent decades. Assessing the impact of sodium intake on gastric cancer will present a more precise and comprehensive assessment of the fact that sodium reduction could serve as the primary prevention approach of the disease. Also, in this study we used region-specific data on sodium intake and the magnitude of the association between sodium intake and gastric cancer to assess the indicator integrating the population-attributable fraction (PAF), which is an estimation of the proportion of gastric cancer cases that could be explained by sodium intake above the WHO recommendation (≤2 g/d).

Given high intakes of sodium and ascending trends observed at a national level, assessing the effect of exposure to high sodium intake could provide integrated knowledge of sodium reduction, serving as the primary prevention approach. To the best of our knowledge, sodium intake on gastric cancer burden has not been systematically evaluated in China. Most importantly, the dynamic trends of gastric cancer burden attributable to high sodium consumption are still unclear since it is necessary to formulate precise prevention strategies. Therefore, we conducted a thorough investigation using the GBD Study 2019, a multinational collaborative and worldwide-updated epidemiological research study, to estimate gastric cancer burden associated with high sodium consumption in China. Targeted activities for the prevention and control of gastric cancer could be achieved by understanding these features.

## 2. Materials and Methods

### 2.1. Study Data

We employed a database produced by the China National Center for Food Safety Risk Assessment from January 1990 to December 2019 to estimate national details, including the consumption of sodium and the incidence and mortality of gastric cancer across China. Also, a database at the regional level monitoring the nutrition and health status of residents in 33 regions was obtained to quantify the estimates of gastric cancer disease burden. We also obtained the global burden of disease in 2019 via the Global Health Data Exchange query tool. Previous studies have described GBD research methods [12]. We acquired data on the annual number of deaths, DALYs, age-standardized mortality rate (ASMR), and age-standardized DALY rate (ASDR) of gastric cancer being attributable to high sodium intake separated by gender, age, regions, and socio-demographic index (SDI) quintile from 1990 to 2019. The gender, age, and regional distribution profiles of gastric cancer disease burden in 33 regions (22 provinces, 4 cities, 5 autonomous regions and 2 special administrative regions) in China were further analyzed. The results were presented as numbers and 95% uncertainty intervals (UIs).

SDI is an aggregative indicator that is calculated by integrating with the lag-distributed income per capita, with educated experience for those age > 15 and above to reflect the economic status of certain geographical locations [13]. Given the data on the global burden of disease in 2017, the SDI in China was 0.71 and the level ranged from 0.47 to 0.86. Specifically, the details could be categorized into four levels; that is, high SDI region (Hong Kong, Macao, Beijing, Shanghai, and Taiwan), high-middle SDI region (Fujian, Guangdong, Hebei, Heilongjiang, Inner Mongolia, Jiangsu, Jilin, Liaoning, Shandong, Tianjin, and Zhejiang), low-middle SDI region (Gansu, Guizhou, and Tibet), and middle SDI region (other regions) [14].

The database is a public domain and does not contain individual information. Therefore, it is unnecessary to contact a Research Ethics Committee for approval.

### 2.2. Definitions of Gastric Cancer and High Sodium Intake Exposure

The classification of gastric cancer in this study is encoded using the 10th edition of the International Statistical Classification of Diseases and Related Health Issues (ICD), with ICD-10, C16 [15]. An average number of urinary excretions of sodium within 24 h (grams per day) >3 g (95% UI: 1 g, 5 g) was considered as the standard definition of high sodium intake [16].

### 2.3. Estimation of High Sodium Intake-Attributed Gastric Cancer Burden

The special approach of the GBD study in 2019 has been elucidated elsewhere previously [17,18,19]. Specifically, attributable deaths, years of life lost (YLLs), years of life lived with disability (YLDs), and disability-adjusted life years (DALYs) for high sodium intake at the regional level separated by country worldwide were estimated based on the GBD 2019 study. The level of sodium consumption in each age, sex, location, and year were assessed through all available data sources, such as the survey representing population and surveillance data using spatio-temporal Gaussian process regression and Disease Modeling Meta regression (DisMod-MR). As an integrative system modeling approach, DisMod-MR is used to quantify the number of deaths separated by age, gender, country, and year in the study.

Deaths were diagnosed as the number of deaths occurring within a certain population in the specific period. YLDs were calculated on standardized disability weights as for health state. And, YLLs were calculated on a reference maximum observed life expectancy. Then, they were summed to calculate DALYs. PAFs (population attributed fractions) were calculated by the exposure level, estimates of relative risk, and theoretical minimum risk level. A comparative assessment approach for risk was used to estimate PAFs of sodium intake outcome pair separated by age, gender, and year [20]. The number of deaths and DALYs being attributable to high sodium intake associated with gastric cancer were estimated by multiplying the PAFs with the number of deaths and DALYs for the specific disease.

The standard equation for the *PAF* is defined as follows [19]:$${PAF}_{asgt}=\frac{\sum_{x=l}^{u}RR_{asg}\left(x\right)P_{asgt}\left(x\right)-RR_{asg}\left({TMREL}_{as}\right)}{\sum_{x=l}^{u}RR_{as}\left(x\right)P_{asgt}\left(x\right)}$$
where *PAF_asgt_* was the *PAF* for gastric cancer burden being attributed to high sodium intake for age group *a*, gender *s*, location *g*, and year *t*. *RR_as__g_* (*x*) was the relative risks between exposure level *x* (from *l* to *u*) of sodium consumption and gastric cancer separated by age group *a*, gender *s*, and year *t*; *P_asgt_* (*x*) was the proportion of the population exposed to sodium intake at the level *x* age group *a*, gender *s*, location *g*, and year *t*. *TMREL_as_* is the *TMREL* regarding age group *a* and gender *s*.

### 2.4. Statistical Analyses

Data on deaths, DALYs, ASMR, and ASDR were presented as numbers with 95% UIs on account of the 2.5th and 97.5th percentiles of the above 1000 estimations [20]. Age was categorized into specific age groups per 5 years (18 subgroups). The regression equation ln (ASR) = α + βx + ε, where x stands for the year, was used to fit the ASR. Then, using the model 100 × (exp (β) − 1) and its 95% CI [21], ASR, including ASMR and ASDR, was regarded as increasing if the lower limit of the 95% CI of the EAPC estimation was larger than zero. Comparatively, the ASR indicated a declining trend if the upper limit of the 95% CI of the EAPC was less than zero. If the 95% CI included zero, the ASR was considered to be steady. Smoothing spline models were determined by the association between the SDI and the EACP in ASMR or ASDR of gastric cancer being attributed to high sodium consumption among 33 regions. As an index reflecting the trends of age-standardized rates (ASR) within a certain period, the estimated annual percentage change (EAPC) was employed to explore the longitudinal trends of the ASMR and ASDR of high sodium intake-related gastric cancer in 1990–2019 [21]. In this study, we employed the Spearman rank test to examine the associations between the ASMR or ASDR of 1990 and the SDI of 2019. R software (version 4.1.2) was employed to perform all statistical analyses. A two-sided *p* value of less than 0.05 was regarded as significant.

## 3. Results

### 3.1. Burden of Gastric Cancer Attributable to High Sodium Intake from 1990 to 2019

In 2019, the number of deaths and DALYs of gastric cancer being attributed to high sodium intake were estimated at 74,098.60 and 1,735,810.69, representing 7.71% (0.23% to 30.82%) and 7.79% (0.22% to 30.89%), respectively, across the world. In China, the numbers were estimated at 37,131.48 and 873,813.19, representing 8.75% (0.21% to 32.76%) and 8.85% (0.21% to 32.82%), respectively (Table 1 and Table 2). In 1990–2019, the number of these indicators increased by one third in males, while a moderate increase was detected in females. The male-to-female ratio was both approximately 2.5 (Table 1 and Table 2).

In China, the high sodium intake-attributed ASMR for gastric cancer declined from 2.28 (95% UI: 0.05 to 8.60) per 100,000 population in 1990 to 1.05 (95% UI: 0.02 to 4.06) in 2019 for females; meanwhile, it decreased from 4.60 (95% UI: 0.10 to 17.09) in 1990 to 2.92 (95% UI: 0.06 to 10.90) in 2019 for males. The ASDR decreased from 54.40 (95% UI: 1.23 to 203.57) per 100,000 population in 1990 to 22.79 (95% UI: 0.50 to 88.76) in 2019 for females; similarly, it decreased from 108.37 (95% UI: 2.36 to 403.47) in 1990 to 63.79 (95% UI: 1.38 to 237.25) in 2019 for males.

Being separated by regions, the EAPC of gastric cancer being attributed to high salt intake in 1990–2019 is shown in Table 1 and Table 2. The ASMR presented a downward trend in both genders (males: −1.22%, 95% CI: −1.60% to −0.83%; females: −2.66%, 95% CI: −3.05% to −2.27%). The ASDR of high sodium intake-related gastric cancer also showed a declining trend (males: −1.49%, 95% CI: 0.17% to 0.32%; females: 3.07%, 95% CI: −3.44% to −2.69%). Interestingly, both China and the global gastric cancer-related ASDR slightly increased, with an EAPC of 0.22% (95% CI: 0.06% to 0.36%) and 0.36% (95% CI: 0.08% to 0.68%), respectively. China and the global gastric cancer-related ASMR had a downward trend, with an EAPC of −1.83% (95% CI: −2.02% to −1.65%) and −1.72% (95% CI: −2.11% to −1.33%), respectively.

### 3.2. Burden of Gastric Cancer Being Attributed to High Sodium Intake Separated by Age and Gender

Age-specific numbers of death and DALYs for gastric cancer being attributed to high sodium intake for both males and females demonstrated an increasing trend with age (*p*_trend_ < 0.05). The number of gastric cancer deaths being attributed to high sodium intake in both genders peaked in the 70–74 age group in 2019 (Figure 1A). The gastric cancer-related DALYs reached their peak in both genders among the population aged 65–69, and these trends were similar for both genders (Figure 1B). The numbers of deaths and DALYs were concentrated in those individuals aged 50–74 and 45–74, respectively. Moreover, in the 25–89 age group, the number of deaths and DALYs for males transcended those of females, whereas for individuals over 90, the number of deaths and DALYs for females transcended those of males (Figure 1A,B).

### 3.3. Burden of Gastric Cancer Attributable to High Sodium Intake Separated by Regions and SDI

As for the regional level, interestingly, the range of PAFs in each region across China is almost similar. Gansu (deaths: 8.04%, 95% UI: 0.21% to 31.41%; DALYs: 8.39%, 95% UI: 0.21% to 32%) had the lowest burden of gastric cancer-related deaths and DALYs attributable to high sodium consumption, while serious burden was found in Zhejiang (deaths: 9.00%, 95% UI: 0.21% to 33.13%; DALYs: 9.01%, 95% UI: 0.21% to 33.07%). Moreover, the indicators of ASMR and ASDR (per 100,000 population) in Qinghai were the highest (ASMR: 4.36, 95% UI: 0.10 to 16.29; ASDR: 101.76, 95% UI: 2.17 to 381.28) and also in Ningxia (ASMR: 3.41, 95% UI: 0.08 to 13.20; ASDR: 74.75, 95% UI: 1.67 to 290.15). In contrast, Macao (ASMR: 0.63, 95% UI: 0.29 to 55.28; ASDR: 14.20, 95% UI: 0.29 to 55.28), Hong Kong (ASMR: 0.72, 95% UI: 0.02 to 2.72; ASDR: 15.34, 95% UI: 0.32 to 58.16), and Beijing (ASMR: 0.78, 95% UI: 0.02 to 2.85; ASDR: 8.92, 95% UI: 0.21 to 32.87) had the lowest indicators (Appendix A and Figure 2).

At the SDI regional level, the largest numbers of deaths and DALYs of gastric cancer being attributed to high sodium consumption in both 1990 and 2019 were detected in the middle SDI regions, while the lowest indicators were detected in the high SDI regions. Similarly, the middle SDI regions had the largest ASMR and ASDR of gastric cancer being attributed to high sodium consumption in both 1990 and 2019, whereas the high SDI regions presented the lowest ASMR and ASDR. Meanwhile, EACP in ASMR and ASDR varied by the SDI quintiles. The ASMR in all SDI regions presented a downward trend. Regarding gastric cancer ASDR being attributed to high sodium intake, the high, high-middle SDI, and middle SDI regions all displayed significant downward trends in 1990–2019 (*p*_trend_ < 0.05). And, the low-middle SDI regions displayed a slight downward trend (*p*_trend_ > 0.05) (shown in Appendix A).

### 3.4. Temporal Trends of Mortality and DALYs of Gastric Cancer Being Attributed to High Sodium Intake from 1990 to 2019

In 1990–2019, the estimated annual percentage change (EAPC) in the ASMR or ASDR of gastric cancer being attributed to high sodium consumption varied obviously among different regions. Shanghai presented the largest drop in the ASMR and ASDR of gastric cancer being attributed to high sodium intake, followed by Beijing, Jilin, and Macao. Additionally, the changing trends in ASMR and ASDR in 1990–2019 were similar across the regions. Shanghai displayed the most significant annual decline, with an EAPC of −3.34% (95% CI: −3.43% to −3.25%) in ASMR and an EAPC of −3.60% (95% CI: −3.70% to −3.49%) in ASDR (shown in Appendix A).

Generally, the DALYs rate at the national level and global level were relatively stable, and EAPCs in DALYs rate slightly decreased from 1990 to 2019 in all regions across China (*p*_trend_ > 0.05) (Figure 3A). Specifically, in the high SDI regions, there was a downward trend of age-specific DALYs in individuals ≤90 and upward in individuals >90 from 1990 to 2019, among which the biggest increase and drop occurred in individuals aged 40–44 and 90–94, respectively. In the low-middle SDI regions, there was a downward trend of the age-specific DALYs rate in individuals ≤80 and upward in individuals aged 80 over, with the biggest increase and decrease in individuals aged 85–90 and 45–50. A similar trend was detected for EAPC in the mortality rate (Figure 3B).

### 3.5. Temporal Trends of Mortality and DALYs of Gastric Cancer Being Attributed to High Sodium Intake in 1990–2019

The ASMR and ASDR of gastric cancer being attributed to high sodium consumption had linear associations with the SDI from 1990 to 2019. The ASMR and ASDR decreased with the increase in SDI quintile. Despite the declining trends of ASMR and ASDR in Shanghai, Jiangsu, Shanxi, Fujian, and Sichuan, these regions still presented a higher ASMR and ASDR than the expected values for the SDI in Beijing, Heilongjiang, Macao, Inner Mongolia, Hubei, Jiangxi, Tianjin, and Hunan. The estimates of gastric cancer-related disease burden were relatively steady and lower than the expected values from 1990 to 2019 (Figure 4A,B).

The baseline disease reservoir was reflected in the ASMR or ASDR of gastric cancer attributable to high sodium intake in 1990. The SDI in 2019 functioned as a standard for the development of each region. There was no association between EAPC in ASMR and ASDR in 1990 (ρ = 0.047, *p* = 0.796); also, there was no association between EAPC in ASMR and ASDR in 1990 (ρ = −0.008, *p* = 0.965) (Figure 5A,B). Figure 5C,D suggested that there was a strong negative association between the EAPC in ASMR and the SDI in 2019 (ρ = −0.642, *p* < 0.001). The EAPC in ASDR and the SDI also exhibited a negative connection in 2019 (ρ = −0.538, *p* = 0.0012).

## 4. Discussion

China contributes the largest numbers of gastric cancer deaths and DALYs worldwide [22]. With the implementation of public health strategies and the changing of lifestyles, potential risk factors (such as smoking, alcohol consumption, a high sodium dietary habit, and H. pylori infection) have been efficiently controlled, thereby lessening the mortality rates and DALYs in China [23]. However, gastric cancer represents one of the most common cancers and serves as the leading cause of cancer deaths. The disease burden of gastric cancer in China still remains serious and the rates are still higher than the average levels across the world. High salt consumption ranks as one of the leading causes of large amounts of non-communicable diseases, including gastric cancer. Given the sodium diet and the observed patterns of exposure to them, we introduced the first large-scale effort to quantify a comprehensive estimate of the spatial profiles and temporal trends of the high sodium intake-related gastric cancer burden from 1990 to 2019.

In our study, the absolute death numbers and DALYs in gastric cancer being attributed to high sodium intake were 37.13 thousand and 873.81 thousand in 2019, increasing one third and one fifth compared with that in 1990. Our study covered a longitudinal period of about 30 years for the risk-attributable disease burden in Chinese residents based on 33 region-level administrative units. The study could provide valuable scientific and theoretical evidence and bring thoughtful insights for preventing gastric cancer at a low cost.

We found that the age-specific mortality and DALY rates of gastric cancer attributable to sodium intakes are higher in males than in females across all age groups. Our findings are consistent with previous studies [16]. Males could excrete more salt through urine (247 mmol/d) than females (218 mmol/d) [24], indicating that males were more likely to be exposed to higher sodium levels than females for the internal environment and thus suffer from more serious disease burdens. Generally, males are prone to being involved in negative lifestyle choices, such as smoking and alcohol abuse [25]. Smoking and alcohol consumption are well-known behavioral factors that are associated with unhealthy dietary patterns for gastric cancer risk. Previously, one study showed that smokers or alcohol drinkers prefer salty food. Smoking and alcohol consumption, as carcinogenic factors, could also exacerbate gastric cancer development when combined with unhealthy dietary habits. Especially, the production of inflammatory markers and the generation of oxidative stress substances, such as oxygen radical species, have been recognized as one of the mechanisms of alcohol consumption that affects the onset and development of gastric cancer [26]. Moreover, females were considerably less susceptible to the negative effects of high sodium intake exposure since females are more likely to effectively maintain Na^+^ homeostasis during acclimatization to the challenges of high-salt dietary patterns [27]. Therefore, gender-based sodium reduction policies seem to be more crucial to prevent the occurrence of gastric cancer.

We also explored the fact that age-specific mortality rates and DALYs rates elevated with age in both genders in 1990 and 2019. It revealed that the burden of gastric cancer caused by higher sodium intake is greater in the elderly than the youngsters. High salt intake in this early stage of life may predispose the stomach to disruption and accelerate cancer development in later life [28]. The stomach is highly susceptible to dietary abuse since the gut undergoes significant cellular and structural changes. Conditioning to salt in early life may affect taste bud sensitivity and influence the need and tolerance for salt in later life. It takes longer to develop gastric cancer for a sequence of preneoplastic lesions in older ages. Prolonged exposure to high sodium intake could potentially increase the susceptibility to gastric cancer among the elderly compared to the young population in this time point. Population aging could also be a great influential factor on the burden of many diseases, including the burden of gastric cancer being attributed to high-salt diets [29]. China has implemented effective sodium reduction measures since 2007, such as accelerating the modification of the general rules of nutrition labeling of prepackaged foods and normalizing nutrition and health science education activities, and this may decrease the disease burden. Consequently, the average sodium daily intake of Chinese residents decreased from 5.9 g in 2002 to 5.2 g in 2012 [30]. Although the salt intake of residents showed a downward trend, it was still 75% higher than the 6 g/d recommended by the Chinese Dietary Guidelines (2016) [31]. Most importantly, the decreasing burden of gastric cancer being attributed to high sodium intake may not compensate the increasingly serious effects on the population day by day. Therefore, the approaches needed to reduce intake will require intensive education and training to prepare low-salt alternatives. Individuals who have a preference for salty foods need to have education regarding appropriate sodium intake and a low-sodium diet should be encouraged, especially among seniors.

Our results showed variations in the age-standardized mortality and DALY rates in 1990–2019 at the regional levels. There were considerable variations in the attributable burden of high sodium intake across regions. The disease burden in Qinghai and Shandong in 2019 were, relatively, the highest. This could be due to the low SES there, which is a characteristic concerning a wide part of population. In particular, in low-income and middle-income regions, sodium intake is more diffuse, diverse, and higher compared to those high-income regions [28]. The local residents in Shandong, one region close to the Yellow River Delta, are prone to consuming seafoods and pickled foods with more sodium ingredients. Additionally, eating out has become more popular than ever before and restaurant dishes often contain high salt. Also, the consumption of prepackaged food continues to increase and increases the exposure to high sodium [32], whereas lower disease burdens were detected in Hong Kong and Macao across China. This may be related to the lighter diet of residents in Hong Kong and Macao. The populations of these areas consume less processed meat and pickled foods since the residents pay more attention to dietary patterns. Lifestyle patterns, alcohol drinking, and a high fat diet could boost the effects of high sodium intake on gastric cancer risk and exert mediating effects. Diminishing the sodium intake could also scale back and lower the susceptibility to gastric cancer due to the emerging interaction effect. Dietary preference in daily life and demographic variables, including expanded life expectancy and aging, might be involved in the serious disease burden of gastric cancer.

As a unique approach to assess socioeconomic status, the SDI was identified as a key factor affecting the mortality and DALY rate of disease [33]. Following the marked differences in sodium intake identified across regions, high SDI regions presented the lowest ASMR and ASDR in 1990–2019. This indicates that residents in high SDI regions have easy access to advanced education and better healthcare, as well as concerted preventative efforts. Most importantly, the ASMR and ASDR of medium SDI areas have surpassed those of low-middle SDI and low SDI regions. Socio-economic status has been associated with variations in morbidity and mortality of the disease among different regions [34]. Lifestyle factors are usually considered as the mediators between social-economic status and health [35]. In this study, EAPC in ASMR of gastric cancer attributable to high sodium intake and SDI quantiles were negatively correlated. The burden of high sodium intake-related gastric cancer in those less-developed regions is significantly higher than that in more developed regions, and the gap is getting larger from 1990 to 2019. Recently, Chinese individual’s average sodium intake has been declining, which could contribute to an obvious drop in the ASMR and ASDR of gastric cancer being attributed to high sodium intake [36]. Reducing sodium intake is still urgent across China, particularly in low SDI regions with high sodium intake diet. Concerted efforts to diminish sodium consumption among all populations are needed to further decrease the gastric cancer burden.

It is critical to explore and further identify the association between sodium intake and health outcome by utilizing an appropriate and efficient assessment tool of sodium level. In our study, urine collection within 24 h was used to assess dietary sodium intake for individuals instead of spot and short-duration timed urine collection methods. Besides these strengths, there are still some limitations that need to be considered. First, the profile of disease burden related to high sodium intake has not been estimated and examined separately for urban and rural regions because of data unavailability. Second, in our study, smoking or alcohol drinking status was not examined and analyzed in-depth, and statistical significance could not be confirmed due to the availability of data. A further limitation was that only individuals aged 25 and above were included in our study, while the details of residents aged ≤ 24 are unavailable. It is uneasy to quantify the level of sodium intake since it is the main component of salt. In the study, 24 h urine samples were used to estimate the dietary intake of sodium, and this might introduce potential biases. Most importantly, details of the association between sodium intake and gastric cancer regarding tumor location or histological type have not been explored. Correspondingly, accurate estimates of the contribution of the gastric cancer burden being attributed to high sodium intake could not be achieved. And, rural regions could not be examined separately because of data availability. Rural populations with an aging population have exceptionally high salt intakes across their lifetimes [37]. While taking these limitations into account, our results still have large amounts of clinical and public health implications.

## 5. Conclusions

Overall, our study provided information on the importance of high sodium consumption as a key risk factor of gastric cancer based on a large and diverse population across different regions in China, with adequate statistical power of the data. This study also presented a profile of the gastric cancer burden being attributed to high sodium diets among Chinese residents from 1990 to 2019. Despite the fact that a decrease in the ASMR and ASDR of gastric cancer being attributed to high sodium intake is detected, the numbers of death and DALYs are still high in 2019. Societies and individuals could succeed in lowering gastric cancer risk by reducing dietary salt consumption. Tailored flexible, integrated, gender-based, and geographic-specific salt reduction programs are anticipated to lesson gastric cancer risk in the future.

## Figures and Tables

**Figure 1 nutrients-15-05088-f001:**
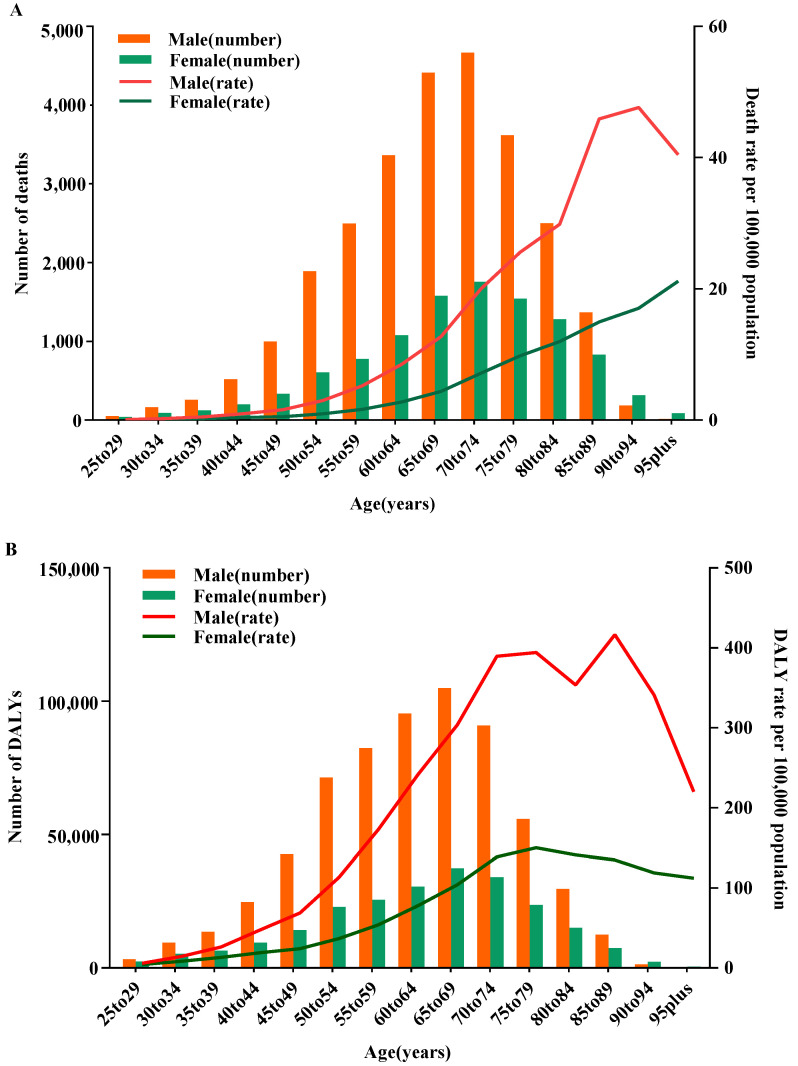
Age-specific numbers and rates of gastric cancer deaths and DALYs being attributed to high sodium consumption by sex, in 2019. (**A**) Deaths. (**B**) DALYs. DALYs, disability-adjusted life years.

**Figure 2 nutrients-15-05088-f002:**
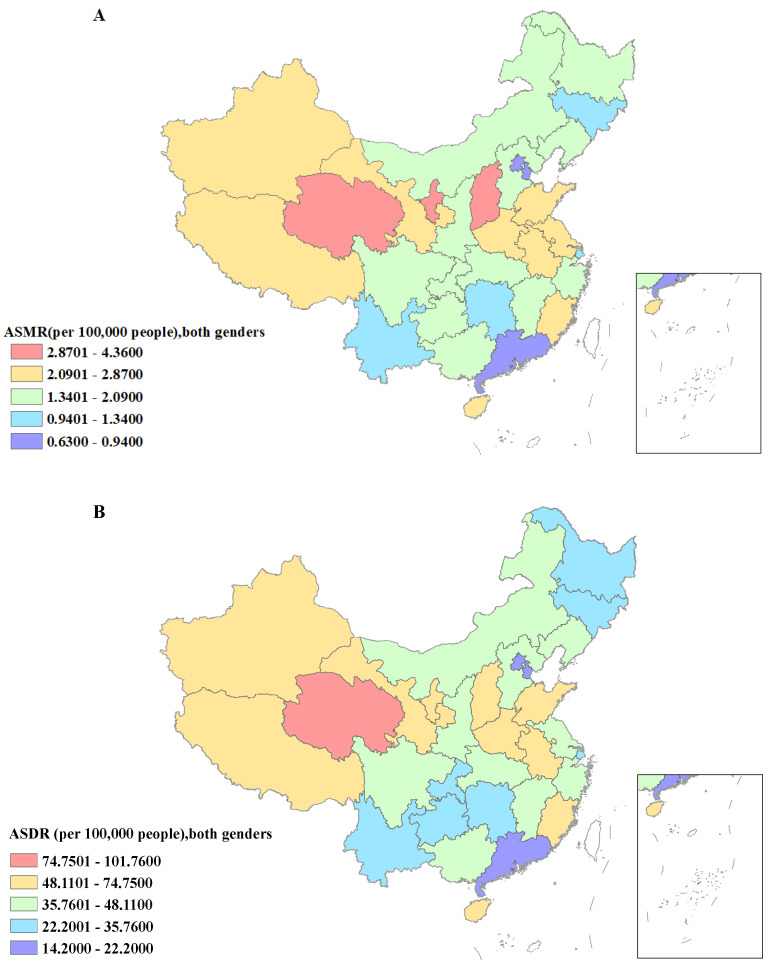
The disease burden of gastric cancer being attributed to high sodium intake for both genders in China. (**A**) The spatial distribution of gastric cancer ASMR being attributed to high sodium intake in 2019. (**B**) The spatial distribution of ASDR in 2019. (**C**) The EAPC in gastric cancer ASMR being attributed to high sodium intake in 1990–2019. (**D**) The EAPC in gastric cancer ASDR in 1990–2019. ASMR, age-standardized mortality rate; DALYs, disability-adjusted life years; ASDR, age-standardized DALYs rate; EAPC, estimated annual percentage.

**Figure 3 nutrients-15-05088-f003:**
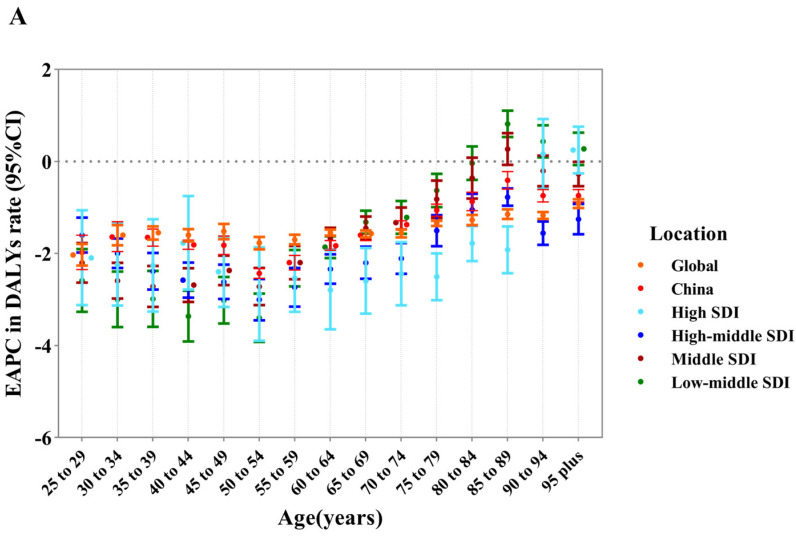
The age distribution in gastric cancer-related mortality rate and DALYs rate being attributed to high sodium intake in 1990−2019 by location. (**A**) EAPC in mortality rate. (**B**) EAPC in DALYs rate.

**Figure 4 nutrients-15-05088-f004:**
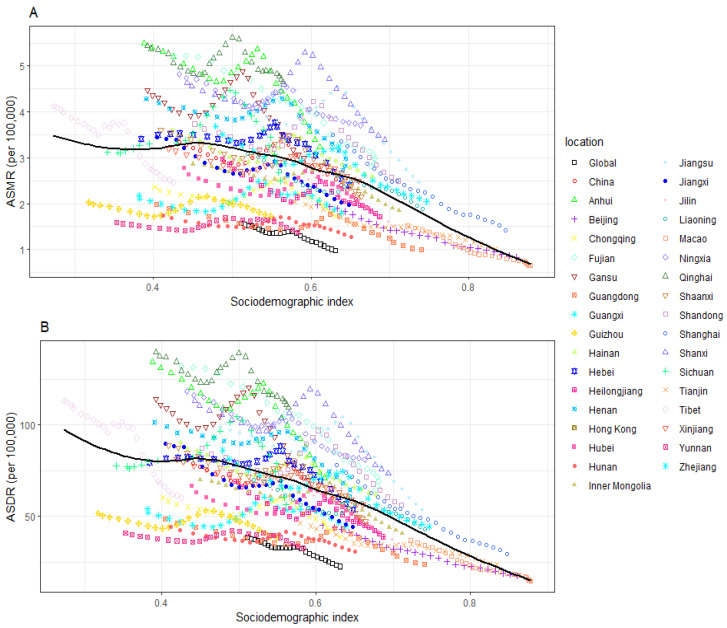
Gastric cancer-related ASMR and ASDR being attributed to high sodium intake across 34 regions by SDI for both genders combined, 1990-2019. For each region, right points depict estimates from each year in 1990-2019. (**A**) The correlation between gastric cancer-related ASMR being attributed to high sodium intake and SDI. (**B**) The correlation between gastric cancer-related ASDR being attributed to high sodium intake and SDI.

**Figure 5 nutrients-15-05088-f005:**
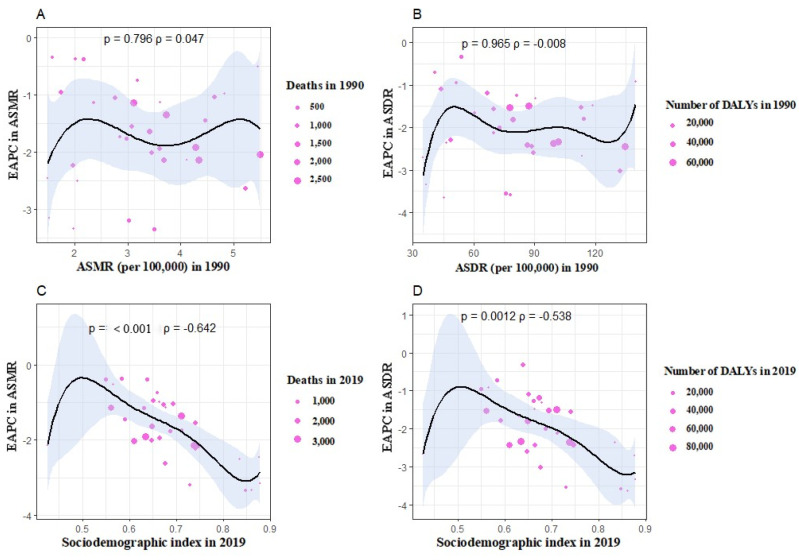
The influential factors for EAPC. (**A**) The correlation between EAPC in ASMR and ASDR in 1990. (**B**) The correlation between EAPC in ASMR and ASDR in 1990. (**C**) The correlation between EAPC in ASMR and SDI in 2019. (**D**) The correlation between EAPC in ASDR and SDI in 2019.

**Table 1 nutrients-15-05088-t001:** Deaths and ASMR of high sodium intake-attributed gastric cancer disease and temporal trends in 1990–2019.

Characteristics	1990		2019			1990–2019
	Deaths Cases, No. (95% UI)	ASMR per 100,000 No. (95% UI)	Deaths Cases, No. (95% UI)	ASMR per 100,000 No. (95% UI)	PAFs % (95% UI)	EAPC(%) in ASMR No. (95% CI)
Global	60,961.34 (1739.22 to 242,439.96)	1.57 (0.05 to 6.26)	74,098.60 (2117.22 to 294,837.24)	0.92 (0.03 to 3.64)	7.71 (0.23 to 30.82)	−1.83 (−2.02 to −1.65)
China	27,226.5 (612.93 to 101,649.43)	3.34 (0.08 to 12.56)	37,131.48 (833.14 to 138,478.72)	1.90 (0.04 to 7.12)	8.75 (0.21 to 32.76)	−1.72 (−2.11 to −1.33)
Gender						
Male	17,739.88 (383.80 to 66,070.45)	4.60 (0.10 to 17.09)	26,497.45 (569.28 to 98,942.04)	2.92 (0.06 to 10.90)	8.84 (0.21 to 0.32.76)	−1.22 (−1.60 to −0.83)
Female	9486.62 (217.09 to 35,696.85)	2.28 (0.05 to 8.60)	10,634.03 (238.25 to 41,305.44)	1.05 (0.02 to 4.06)	8.53 (0.21 to 0.32.36)	−2.66 (−3.05 to −2.27)
Socio-demographic Index (SDI)						
High-middle SDI	11,812.80 (260.38 to 44,237.97)	3.30 (0.07 to 12.46)	16,331.75 (356.21 to 60,601.55)	1.73 (0.04 to 6.46)	8.75 (0.21 to 32.72)	−2.08 (−2.43 to −1.73)
High SDI	86.11 (2.01 to 315.06)	1.505 (0.03 to 5.49)	113.07 (2.41 to 432.47)	0.675 (0.02 to 2.57)	8.81 (0.21 to 32.93)	−2.80 (−2.9 to −2.59)
Low-middle SDI	1086.65 (24.60 to 4145.30)	3.54 (0.08 to 13.62)	1710.31 (38.54 to 6503.53)	2.25 (0.05 to 8.52)	8.55 (0.21 to 32.32)	−1.32 (−1.66 to −0.97)
Middle SDI	14,240.96 (313.44 to 52,961.04)	3.40 (0.08 to 12.79)	18,976.33 (411.09 to 71,070.87)	2.12 (0.05 to 8.03)	8.72 (0.21 to 32.66)	−1.41 (−1.74 to −1.03)

**Table 2 nutrients-15-05088-t002:** DALYs and ASDR of high sodium intake-attributed gastric cancer disease and temporal trends in 1990–2019.

Characteristics	1990		2019			1990–2019
	DALYs, No. (95% UI)	ASDR per 100,000 No. (95% UI)	DALYs, No. (95% UI)	ASDR per 100,000 No. (95% UI)	PAFs % (95% UI)	EAPC (%) in ASDR No. (95% CI)
Global	1,598,735.69 (44,382.75 to 6,299,148.27)	38.52 (1.08 to 152.39)	1,735,810.69 (48,674.79 to 6,804,671.03)	20.91 (0.59 to 82.11)	7.79 (0.22 to 30.89)	0.22 (0.06 to 0.36)
China	734,447.94 (16,388.16 to 2,731,936.05)	80.72 (1.81 to 301.47)	873,813.19 (19,283.13 to 3,220,231.82)	42.52 (0.94 to 157.03)	8.85 (0.21 to 32.82)	0.36 (0.08 to 0.68)
Gender						
Male	485,849.57 (10,416.51 to 1,808,630.53)	108.37 (2.36 to 403.47)	637,200.70 (13,603.95 to 2,377,631.68)	63.79 (1.38 to 237.25)	8.91 (0.21 to 32.96)	−1.49 (−1.90 to −1.09)
Female	248,598.36 (5605.48 to 927,208.28)	54.40 (1.23 to 203.57)	236,612.49 (5160.81 to 921,927.87)	22.79 (0.50 to 88.76)	8.68 (0.21 to 32.51)	−3.07 (−3.44 to −2.69)
Socio-demographic Index (SDI)						
High-middle SDI	310,958.47 (6754.15 to 1,163,118.97)	78.83 (1.73 to 296.67)	383,445.35(8214.15 to 1,420,311.21)	38.88 (0.84 to 144.20)	8.85 (0.21 to 32.85)	−2.27 (−2.64 to −1.89)
High SDI	2041.38 (47.30 to 7501.81)	18.24 (0.42 to 66.82)	2100.07(543.96 to 7986.59)	7.99 (0.17 to 30.29)	8.84 (0.21 to 32.91)	−2.925 (−3.15 to −3.15)
Low-middle SDI	185,247.99 (5180.19 to 751,627.37)	27.61 (0.78 to 112.47)	280,424.65 (8018.74 to 1,137,545.52)	19.28 (0.55 to 78.31)	7.42 (0.22 to 30.20)	0.65 (0.47 to 0.84)
Middle SDI	389,623.72(8469.43 to 1,447,292.53)	83.72 (1.83 to 313.97)	446,547.11 (9506.97 to 1,666,429.48)	47.82 (1.03 to 179.74)	8.33 (0.21 to 32.78)	−1.77 (−2.08 to −1.45)

## Data Availability

The authors declare that the data of this study are available upon reasonable request.

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
