# Peer review of "The Burden of Gastric Cancer Attributable to High Sodium Intake: A Longitudinal Study from 1990 to 2019 in China"

_nutrients, 2023, doi:10.3390/nu15245088_

Round 1
Reviewer 1 Report
Comments and Suggestions for Authors
This study employed GBD data and data from the China National Center for Food Safety Risk Assessment to investigate the gastric cancer burden with DALY, PAF, ASMR, etc. Some specific problems have to be addressed before it can be published.
1. There are conflicted statements in terms of data usage. On page 3, it states that the database is a public domain and does not contain individual information. Therefore, it is unnecessary to obtain IRB approval. But on page 20, in the IRB statement, it states that all participants provided written informed consent. It casts doubts about how the data were actually acquired.
2. In GBD data, there is no regional or local data within China. So, most of the data employed in this study is assumed to be from the China National Center for Food Safety Risk Assessment. However, there is no explanation or introduction about what the data is like and how they were collected. It is basically a black box.
3. The formula of PAF on page 4 is out of place and cannot be read fully.
4. The policy of sodium reduction measures implemented in 2007 should be elaborated. This policy should affect all ages, not just the specific age groups.
5. The elderly have a higher death rate than the younger cohorts. It would be caused by more prolonged exposure to high sodium intake or due to the natural cause of aging, which should be discussed further.
6. Many abbreviations were not used in the full term at first time, especially in the abstract, such as GBD and SDI.
Comments on the Quality of English Language
English writing needs significant editing. There are grammatical mistakes, and some paragraphs do not clearly express the ideas—for example, the 2nd paragraph of page 19.
Author Response
For research article
Response to Reviewer X Comments |
||
1. Summary |
|
|
Thank you very much for taking the time to review this manuscript. Please find the detailed responses below and the corresponding revisions/corrections highlighted/in track changes in the re-submitted files. |
||
2. Questions for General Evaluation |
Reviewer’s Evaluation |
Response and Revisions |
Does the introduction provide sufficient background and include all relevant references? |
Yes |
|
Are all the cited references relevant to the research? |
Yes |
|
Is the research design appropriate? |
Can be improved |
We have made some improvements. |
Are the methods adequately described? |
Must be improved |
We have made it clear. |
Are the results clearly presented? |
Must be improved |
We have made some changes. |
Are the conclusions supported by the results? |
Must be improved |
We have made some modifications. |
3. Point-by-point response to Comments and Suggestions for Authors |
||
Comments 1: There are conflicted statements in terms of data usage. On page 3, it states that the database is a public domain and does not contain individual information. Therefore, it is unnecessary to obtain IRB approval. But on page 20, in the IRB statement, it states that all participants provided written informed consent. It casts doubts about how the data were actually acquired. |
||
Response:Thank you for pointing this out. We absolutely agree with this wonderful comment. Therefore, we have deleted the related content on page 3. All details have been presented on page 11, line 493.
|
||
Comments 2: In GBD data, there is no regional or local data within China. So, most of the data employed in this study is assumed to be from the China National Center for Food Safety Risk Assessment. However, there is no explanation or introduction about what the data is like and how they were collected. It is basically a black box. |
||
Response: Agree. Thank you for your valuable suggestions. We have, accordingly, added details to emphasize this point. The content is as follows: ”We used national estimates of the details, including the consumption of sodium intake and the incidence and mortality of gastric cancer across China, a database produced by the China National Center for Food Safety Risk Assessment from January 1990 to December 2019. Database at the level of regions monitoring the nutrition and health status of residents in 33 provinces were obtained to quantify the estimates of gastric cancer disease burden. ” – page 3, line 113 to 118.
|
||
Comments 3: The formula of PAF on page 4 is out of place and cannot be read fully. |
||
Response: Agree. Thank you for your valuable suggestions. As you suggested, we have retyped the formula. “”– page 4, line 166.
|
||
Comments 4: The policy of sodium reduction measures implemented in 2007 should be elaborated. This policy should affect all ages, not just the specific age groups. |
||
Response: Thank you for pointing this out. We agree with this good comment. Therefore, we have added description of this policy. The content is as follows: ”China has implemented effective sodium reduction measures since 2007, such as accelerating the revision of the general rules of nutrition labeling of prepackaged foods and normalizing nutrition and health science education activities, and this may decrease the disease burden. Consequently, the average sodium daily intake in Chinese residents decreased from 5.9g in 2002 to 5.2g in 2012[30]. Although the salt intake of residents showed a downward trend, it was still 75% higher than the 6g/d recommended by the Chinese Dietary Guidelines (2016)[31]. Most importantly, the decreasing burden of gastric cancer attributable to high sodium intake may not offset the effects of more and more serious population. Therefore, the approaches needed to reduce intake will require intensive education and training to prepare low-salt alternatives. Individuals who have preference for salty foods need to have education regarding appropriate sodium intake and a low-sodium diet should be encouraged, especially among the seniors.”– page 9, line 393 to 405.
|
||
Comments 5: The elderly have a higher death rate than the younger cohorts. It would be caused by more prolonged exposure to high sodium intake or due to the natural cause of aging, which should be discussed further. |
||
Response 5: Thank you for pointing this out. We agree with this good comment. As you suggested, we have made extensive modifications to our manuscript and supplementary tables to make the explanation clear. The content is as follows: "High salt intake at this early stage of life may predispose the stomach to disruption and accelerate cancer development in later life[28]. Stomach is highly susceptible to dietary insults since the gut undergoes significant cellular and structural changes. Conditioning to salt in early life may affect taste bud sensitivity and influence the need and tolerance for salt in later life. It takes longer to develop gastric cancer for a sequence of preneoplastic lesions in older ages. A prolonged exposure to high sodium intake could potentially increase the susceptibility to gastric cancer among the elderly compared to the young population in the special time point. Population aging could also be a great influential factor on the burden of whole diseases, including the burden of gastric cancer attributable to high-salt diets[29]. ”– page 9, line 383 to 393.
|
||
Comments 6: Many abbreviations were not used in the full term at first time, especially in the abstract, such as GBD and SDI. |
||
Response: Thank you for pointing this out. We agree with this comment. Therefore, we have used full term to suggest abbreviations at first time–”GBD(Global Burden of Disease Study) ””SDI(Socio-demographic index)” --page 1, line 22 and line 31.
|
||
4. Response to Comments on the Quality of English Language |
||
Point 1:English writing needs significant editing. There are grammatical mistakes, and some paragraphs do not clearly express the ideas—for example, the 2nd paragraph of page 19. |
||
Response 1: We have tried our best to improve the manuscript and made some critical changes in the manuscript. And here we did not list the changes but marked in red in the revised paper. We appreciate for reviewers’ wonderful work and we hope that the correction and the polish will promote the quality of the manuscript and meet the standard for publishing. |
||
5. Additional clarifications |
||
|
Supplementary Tables 1. Deaths and ASMR of gastric cancer disease attributable-high sodium intake in 1990 and 2019 and the temporal trends from 1990-2019.
Characteristics |
1990 |
2019 |
1990-2019 |
|||
|
Deaths cases, No. (95% UI) |
ASMR per 100,000 No.(95% UI) |
Deaths cases, No. (95% UI) |
ASMR per 100,000 No. (95% UI) |
PAFs % (95% UI) |
EAPC(%) in ASMR No. (95% CI) |
Region |
||||||
Anhui |
2146.88 (46.72 to 7905.63) |
5.49 (0.12 to 20.37) |
2620.63 (55.77 to 9898) |
2.87 (0.06 to 10.85) |
8.89 (0.21 to 33.04) |
-2.03 (-2.44 to -1.63) |
Beijing |
174.73 (3.86 to 651.83) |
1.98 (0.04 to 7.31) |
252.11 (5.41 to 921.33) |
0.78 (0.02 to 2.85) |
8.90 (0.21 to 32.91) |
-3.32 (-3.41 to -3.23) |
Chongqing |
278.69 (6.04 to 1032.87) |
2.36 (0.05 to 8.71) |
643.60 (13.83 to 2461.93) |
1.52 (0.03 to 5.85) |
8.69 (0.21 to 32.50) |
-1.14 (-1.54 to -0.74) |
Fujian |
1042.99 (22.75 to 3940.85) |
5.22 (0.12 to 19.57) |
1234.75 (26.77 to 4643.99) |
2.47 (0.05 to 9.31) |
8.81(0.21 to 32.80) |
-2.63 (-2.72 to -2.54) |
Gansu |
612.58 (14.23 to 2391.08) |
4.46 (0.11 to 17.82) |
975.81 (22.56 to 3759.51) |
2.84 (0.07 to 10.99) |
8.04 (0.21 to 31.41) |
-1.44 (-1.89 to -0.98) |
Guangdong |
890.53 (19.77 to 3437.91) |
1.98 (0.04 to 7.65) |
1215.51 (27.48 to 4638.79) |
0.93 (0.02 to 3.55) |
8.59 (0.21 to 32.40) |
-2.23 (-2.68 to -1.77) |
Guangxi |
621.23 (13.68 to 2397.13) |
2.17 (0.05 to 8.51) |
1003.72 (22.01 to 3781.63) |
1.63 (0.04 to 6.17) |
8.46 (0.21 to 32.10) |
-0.38 (-0.83 to 0.07) |
Guizhou |
413.55 (9.05 to 1525.41) |
2.03 (0.05 to 7.56) |
672.22 (14.67 to 2516.46) |
1.57 (0.03 to 5.96) |
8.68 (0.21 to 32.59) |
-0.38 (-0.71 to -0.05) |
Hainan |
160.62 (3.58 to 615.73) |
3.60 (0.08 to 13.95) |
256.92 (5.75 to 989.52) |
2.30 (0.05 to 8.89) |
8.42 (0.21 to 32.14) |
-1.12 (-1.42 to -0.82) |
Hebei |
1509.07 (33.34 to 5589.76) |
3.41 (0.08 to 12.69) |
2080.59 (44.65 to 7700.39) |
2.04 (0.04 to 7.56) |
8.83 (0.21 to 32.91) |
-1.63 (-2.09 to -1.18) |
Heilongjiang |
589.98 (13.10 to 2252.24) |
2.98 (0.07 to 11.43) |
929.05 (20.52 to 3481.43) |
1.57 (0.04 to 6.02) |
8.52 (0.21 to 32.26) |
-1.76 (-2.05 to -1.46) |
Henan |
2611.86 (57.8 to 9561.77) |
4.28 (0.10 to 15.83) |
2950.71 (64.73 to 10986.83) |
2.41 (0.05 to 9.02) |
8.79 (0.21 to 32.91) |
-1.91 (-2.30 to -1.52) |
Hong Kong |
81.96 (1.92 to 299.89) |
1.49 (0.03 to 5.46) |
107.24 (2.29 to 409.98) |
0.72 (0.02 to 2.72) |
8.79 (0.21 to 32.88) |
-2.46 (-2.69 to -2.22) |
Hubei |
1035.70 (23.33 to 3810.60) |
2.77 (0.06 to 10.18) |
1492.15 (32.15 to 5669.98) |
1.73 (0.04 to 6.54) |
8.72 (0.21 to 32.71) |
-1.06 (-1.44 to -0.68) |
Hunan |
762.78 (17.22 to 2820.83) |
1.75 (0.04 to 6.47) |
1132.43 (24.55 to 4254.50) |
1.16 (0.03 to 4.36) |
8.75 (0.21 to 32.69) |
-0.96 (-1.25 to -0.66) |
Inner Mongolia |
357.90 (7.78 to 1336.06) |
2.86 (0.06 to 10.7) |
574.57 (12.24 to 2104.58) |
1.69 (0.04 to 6.19) |
8.88 (0.21 to 32.98) |
-1.73 (-2.08 to -1.38) |
Jiangsu |
2330.12 (52.07 to 8597.48) |
4.34 (0.10 to 16.1) |
3034.31 (65.72 to 11195.55) |
2.22 (0.05 to 8.17) |
8.82 (0.21 to 32.89) |
-2.14 (-2.65 to -1.62) |
Jiangxi |
846.71 (18.73 to 3142.73) |
3.45 (0.08 to 12.83) |
1016.95 (21.67 to 3815.86) |
1.80 (0.04 to 6.80) |
8.89 (0.21 to 33.01) |
-2.01 (-2.25 to -1.77) |
Jilin |
468.81 (10.31 to 1808.05) |
3.02 (0.07 to 11.66) |
530.16 (11.84 to 1939.54) |
1.25 (0.03 to 4.65) |
8.75 (0.21 to 32.75) |
-3.19 (-3.62 to -2.75) |
Liaoning |
859.80 (19.15 to 3286.72) |
3.08 (0.07 to 11.86) |
1355.75 (29.84 to 5145.03) |
1.76 (0.04 to 6.73) |
8.59 (0.21 to 32.39) |
-1.55 (-1.8 to -1.31) |
Macao |
4.15 (0.09 to 15.17) |
1.52 (0.03 to 5.52) |
5.83 (0.12 to 22.49) |
0.63 (0.01 to 2.42) |
8.83 (0.21 to 32.97) |
-3.14 (-3.34 to -2.95) |
Ningxia |
113.15 (2.47 to 438.24) |
4.82 (0.11 to 18.84) |
243.82 (5.48 to 937.69) |
3.41 (0.08 to 13.20) |
8.44 (0.21 to 32.27) |
-0.98 (-1.13 to -0.82) |
Qinghai |
127.81 (2.69 to 476.30) |
5.44 (0.12 to 20.04) |
273.51 (5.83 to 1019.98) |
4.36 (0.10 to 16.29) |
8.88 (0.21 to 32.97) |
-0.51 (-0.71 to -0.3) |
Shaanxi |
796.28 (17.31 to 2954.11) |
3.61 (0.08 to 13.51) |
1044.35 (22.34 to 3865.56) |
1.98 (0.04 to 7.39) |
8.84 (0.21 to 32.83) |
-1.94 (-2.31 to -1.58) |
Shandong |
2363.43 (51.86 to 8795.77) |
3.73 (0.08 to 14.01) |
3543.69 (78.12 to 13024.72) |
2.31 (0.05 to 8.52) |
8.86 (0.21 to 32.97) |
-1.35 (-1.88 to -0.82) |
Shanghai |
492.02 (10.58 to 1852.30) |
3.50 (0.08 to 13.27) |
566.30 (12.04 to 2139.24) |
1.34 (0.03 to 5.06) |
8.70 (0.21 to 32.66) |
-3.34 (-3.43 to -3.25) |
Shanxi |
926.59 (20.74 to 3533.39) |
4.64 (0.11 to 17.87) |
1494.43 (32.86 to 5639.88) |
3.16 (0.07 to 12.10) |
8.59 (0.21 to 32.38) |
-1.05 (-1.52 to -0.57) |
Sichuan |
2491.59 (54.53 to 9295.86) |
3.11 (0.07 to 11.65) |
2747.61 (59.9 to 10112.62) |
2.09 (0.05 to 7.76) |
8.70 (0.21 to 32.57) |
-1.14(-1.91 to -0.36) |
Tianjin |
144.33 (3.25 to 539.00) |
2.04 (0.05 to 7.69) |
199.11 (4.28 to 773.61) |
0.94 (0.02 to 3.63) |
8.60 (0.21 to 32.48) |
-2.49 (-2.67 to -2.32) |
Tibet |
60.52 (1.32 to 228.81) |
4.12 (0.09 to 15.49) |
62.28 (1.31 to 227.56) |
2.35 (0.05 to 8.60) |
8.92 (0.21 to 32.95) |
-2.13 (-2.37 to -1.89) |
Xinjiang |
269.27 (5.72 to 1019.43) |
3.19 (0.07 to 12.12) |
559.31 (11.97 to 2077.37) |
2.34 (0.05 to 8.76) |
8.88 (0.21 to 32.84) |
-0.75 (-0.96 to -0.54) |
Yunnan |
385.05 (8.44 to 1452.29) |
1.59 (0.04 to 6.00) |
677.78 (14.8 to 2498.95) |
1.24 (0.03 to 4.61) |
8.71 (0.21 to 32.70) |
-0.36 (-0.69 to -0.02) |
Zhejiang |
1255.84 (27 to 4654.13) |
3.68 (0.08 to 13.67) |
1634.26 (34.75 to 5953.92) |
1.85 (0.04 to 6.76) |
9.00 (0.21 to 33.13) |
-2.14 (-2.44 to -1.83) |
ASMR=age-standardized mortality rate; PAF=population attributable fraction; EAPC=estimated annual percentage change.
Supplementary Tables 2. DALYs and ASDR of gastric cancer disease attributable to high sodium intake in 1990 and 2019 and the temporal trends from 1990 to 2019.
Characteristics |
1990 |
2019 |
1990-2019 |
|||
|
Deaths cases, No. (95% UI) |
ASDR per 100,000 No.(95% UI) |
DALYs, No. (95% UI) |
ASDR per 100,000 No. (95% UI) |
PAFs % (95% UI) |
EAPC(%) in ASMR No. (95% CI) |
Region |
||||||
Anhui |
2146.88 (46.72 to 7905.63) |
5.49 (0.12 to 20.37) |
2620.63 (55.77 to 9898) |
2.87 (0.06 to 10.85) |
8.89 (0.21 to 33.04) |
-2.03 (-2.44 to -1.63) |
Beijing |
174.73 (3.86 to 651.83) |
1.98 (0.04 to 7.31) |
252.11 (5.41 to 921.33) |
0.78 (0.02 to 2.85) |
8.90 (0.21 to 32.91) |
-3.32 (-3.41 to -3.23) |
Chongqing |
278.69 (6.04 to 1032.87) |
2.36 (0.05 to 8.71) |
643.60 (13.83 to 2461.93) |
1.52 (0.03 to 5.85) |
8.69 (0.21 to 32.50) |
-1.14 (-1.54 to -0.74) |
Fujian |
1042.99 (22.75 to 3940.85) |
5.22 (0.12 to 19.57) |
1234.75 (26.77 to 4643.99) |
2.47 (0.05 to 9.31) |
8.81(0.21 to 32.80) |
-2.63 (-2.72 to -2.54) |
Gansu |
612.58 (14.23 to 2391.08) |
4.46 (0.11 to 17.82) |
975.81 (22.56 to 3759.51) |
2.84 (0.07 to 10.99) |
8.04 (0.21 to 31.41) |
-1.44 (-1.89 to -0.98) |
Guangdong |
890.53 (19.77 to 3437.91) |
1.98 (0.04 to 7.65) |
1215.51 (27.48 to 4638.79) |
0.93 (0.02 to 3.55) |
8.59 (0.21 to 32.40) |
-2.23 (-2.68 to -1.77) |
Guangxi |
621.23 (13.68 to 2397.13) |
2.17 (0.05 to 8.51) |
1003.72 (22.01 to 3781.63) |
1.63 (0.04 to 6.17) |
8.46 (0.21 to 32.10) |
-0.38 (-0.83 to 0.07) |
Guizhou |
413.55 (9.05 to 1525.41) |
2.03 (0.05 to 7.56) |
672.22 (14.67 to 2516.46) |
1.57 (0.03 to 5.96) |
8.68 (0.21 to 32.59) |
-0.38 (-0.71 to -0.05) |
Hainan |
160.62 (3.58 to 615.73) |
3.60 (0.08 to 13.95) |
256.92 (5.75 to 989.52) |
2.30 (0.05 to 8.89) |
8.42 (0.21 to 32.14) |
-1.12 (-1.42 to -0.82) |
Hebei |
1509.07 (33.34 to 5589.76) |
3.41 (0.08 to 12.69) |
2080.59 (44.65 to 7700.39) |
2.04 (0.04 to 7.56) |
8.83 (0.21 to 32.91) |
-1.63 (-2.09 to -1.18) |
Heilongjiang |
589.98 (13.10 to 2252.24) |
2.98 (0.07 to 11.43) |
929.05 (20.52 to 3481.43) |
1.57 (0.04 to 6.02) |
8.52 (0.21 to 32.26) |
-1.76 (-2.05 to -1.46) |
Henan |
2611.86 (57.8 to 9561.77) |
4.28 (0.10 to 15.83) |
2950.71 (64.73 to 10986.83) |
2.41 (0.05 to 9.02) |
8.79 (0.21 to 32.91) |
-1.91 (-2.30 to -1.52) |
Hong Kong |
81.96 (1.92 to 299.89) |
1.49 (0.03 to 5.46) |
107.24 (2.29 to 409.98) |
0.72 (0.02 to 2.72) |
8.79 (0.21 to 32.88) |
-2.46 (-2.69 to -2.22) |
Hubei |
1035.70 (23.33 to 3810.60) |
2.77 (0.06 to 10.18) |
1492.15 (32.15 to 5669.98) |
1.73 (0.04 to 6.54) |
8.72 (0.21 to 32.71) |
-1.06 (-1.44 to -0.68) |
Hunan |
762.78 (17.22 to 2820.83) |
1.75 (0.04 to 6.47) |
1132.43 (24.55 to 4254.50) |
1.16 (0.03 to 4.36) |
8.75 (0.21 to 32.69) |
-0.96 (-1.25 to -0.66) |
Inner Mongolia |
357.90 (7.78 to 1336.06) |
2.86 (0.06 to 10.7) |
574.57 (12.24 to 2104.58) |
1.69 (0.04 to 6.19) |
8.88 (0.21 to 32.98) |
-1.73 (-2.08 to -1.38) |
Jiangsu |
2330.12 (52.07 to 8597.48) |
4.34 (0.10 to 16.1) |
3034.31 (65.72 to 11195.55) |
2.22 (0.05 to 8.17) |
8.82 (0.21 to 32.89) |
-2.14 (-2.65 to -1.62) |
Jiangxi |
846.71 (18.73 to 3142.73) |
3.45 (0.08 to 12.83) |
1016.95 (21.67 to 3815.86) |
1.80 (0.04 to 6.80) |
8.89 (0.21 to 33.01) |
-2.01 (-2.25 to -1.77) |
Jilin |
468.81 (10.31 to 1808.05) |
3.02 (0.07 to 11.66) |
530.16 (11.84 to 1939.54) |
1.25 (0.03 to 4.65) |
8.75 (0.21 to 32.75) |
-3.19 (-3.62 to -2.75) |
Liaoning |
859.80 (19.15 to 3286.72) |
3.08 (0.07 to 11.86) |
1355.75 (29.84 to 5145.03) |
1.76 (0.04 to 6.73) |
8.59 (0.21 to 32.39) |
-1.55 (-1.8 to -1.31) |
Macao |
4.15 (0.09 to 15.17) |
1.52 (0.03 to 5.52) |
5.83 (0.12 to 22.49) |
0.63 (0.01 to 2.42) |
8.83 (0.21 to 32.97) |
-3.14 (-3.34 to -2.95) |
Ningxia |
113.15 (2.47 to 438.24) |
4.82 (0.11 to 18.84) |
243.82 (5.48 to 937.69) |
3.41 (0.08 to 13.20) |
8.44 (0.21 to 32.27) |
-0.98 (-1.13 to -0.82) |
Qinghai |
127.81 (2.69 to 476.30) |
5.44 (0.12 to 20.04) |
273.51 (5.83 to 1019.98) |
4.36 (0.10 to 16.29) |
8.88 (0.21 to 32.97) |
-0.51 (-0.71 to -0.3) |
Shaanxi |
796.28 (17.31 to 2954.11) |
3.61 (0.08 to 13.51) |
1044.35 (22.34 to 3865.56) |
1.98 (0.04 to 7.39) |
8.84 (0.21 to 32.83) |
-1.94 (-2.31 to -1.58) |
Shandong |
2363.43 (51.86 to 8795.77) |
3.73 (0.08 to 14.01) |
3543.69 (78.12 to 13024.72) |
2.31 (0.05 to 8.52) |
8.86 (0.21 to 32.97) |
-1.35 (-1.88 to -0.82) |
Shanghai |
492.02 (10.58 to 1852.30) |
3.50 (0.08 to 13.27) |
566.30 (12.04 to 2139.24) |
1.34 (0.03 to 5.06) |
8.70 (0.21 to 32.66) |
-3.34 (-3.43 to -3.25) |
Shanxi |
926.59 (20.74 to 3533.39) |
4.64 (0.11 to 17.87) |
1494.43 (32.86 to 5639.88) |
3.16 (0.07 to 12.10) |
8.59 (0.21 to 32.38) |
-1.05 (-1.52 to -0.57) |
Sichuan |
2491.59 (54.53 to 9295.86) |
3.11 (0.07 to 11.65) |
2747.61 (59.9 to 10112.62) |
2.09 (0.05 to 7.76) |
8.70 (0.21 to 32.57) |
-1.14(-1.91 to -0.36) |
Tianjin |
144.33 (3.25 to 539.00) |
2.04 (0.05 to 7.69) |
199.11 (4.28 to 773.61) |
0.94 (0.02 to 3.63) |
8.60 (0.21 to 32.48) |
-2.49 (-2.67 to -2.32) |
Tibet |
60.52 (1.32 to 228.81) |
4.12 (0.09 to 15.49) |
62.28 (1.31 to 227.56) |
2.35 (0.05 to 8.60) |
8.92 (0.21 to 32.95) |
-2.13 (-2.37 to -1.89) |
Xinjiang |
269.27 (5.72 to 1019.43) |
3.19 (0.07 to 12.12) |
559.31 (11.97 to 2077.37) |
2.34 (0.05 to 8.76) |
8.88 (0.21 to 32.84) |
-0.75 (-0.96 to -0.54) |
Yunnan |
385.05 (8.44 to 1452.29) |
1.59 (0.04 to 6.00) |
677.78 (14.8 to 2498.95) |
1.24 (0.03 to 4.61) |
8.71 (0.21 to 32.70) |
-0.36 (-0.69 to -0.02) |
Zhejiang |
1255.84 (27 to 4654.13) |
3.68 (0.08 to 13.67) |
1634.26 (34.75 to 5953.92) |
1.85 (0.04 to 6.76) |
9.00 (0.21 to 33.13) |
-2.14 (-2.44 to -1.83) |
DALYs=disability-adjusted life years; ASDR=age-standardized DALYs rate; PAF=population attributable fraction; EAPC=estimated annual percentage change.

Reviewer 2 Report
Comments and Suggestions for Authors
The researchers have conducted a thorough and detailed analysis of data from the GBD Study 2019, a multinational collaborative and updated worldwide epidemiological research, to estimate how the levels of gastric cancer associated with high sodium intake have altered throughout China between 1990 and 2019. Overall, their findings show that the rates of gastric cancer burden attributable to a high-sodium diet have declined over the study period but remain unacceptably high throughout China with significant regional differences. The study tentatively identifies socio-economic/educational factors, gender differences, and age as factors greatly influencing the rates of gastric cancer burden attributable to a high-sodium diet. It recommends that societies and individuals could lower their risk of gastric cancer by reducing dietary salt intake.
The study provides an excellent data reference for medical and health policymakers in identifying the overall scale of the ongoing problem in China and highlighting regions or demographics which need support. However, there is little or no discussion as to why salt intakes remain intransigently high or why there are significant regional differences.
What proportion of the salt intake is considered added to food after the meal's preparation? What proportion is a constituent of a traditionally prepared meal? If the latter, the approaches needed to reduce intake will require intensive education and training to prepare low-salt alternatives. The incidences increase with ageing. So, do rural populations with an ageing population have exceptionally high salt intakes (present and lifetime) and high gastric cancer rates?
The authors indicate no data on salt intake or gastric cancer for adolescents (<25 yr). See Li et al, 2022 and Hagag et al, 2022.Acquiring this information would be particularly important because the gut undergoes significant cellular and structural changes and is highly susceptible to dietary insults. High salt intake at this early stage of life may predispose the stomach to disruption and cancer in later life. Also, conditioning to salt in early life may affect taste bud sensitivity and the need/tolerance for salt in the diet in later life. Urbanization of the population is a significant change in society. Is there any indication that this is affecting the levels of salt intake or gastric cancer?
Ln 17 “J:[email protected]) Is this meant to be here?
Ln 71-72 “A healthy diet and lifestyle are required, especially taste preference and dietary behaviors.” Is this sentence necessary?
Ln 89 Accepting that high salt intake is the primary factor, add a paragraph on other factors that may contribute to or amplify the effects of sodium.
Ln 100 “s.”. Delete.
Ln 127-128 “High sodium intake was defined as an average 24-h urinary sodium excretion (grams per day) greater than 3g (95%UI:1g,5g) [16].”
An average 24-hour urinary sodium excretion (grams per day) of greater than 3g (95%UI:1g,5g) was used as a clear indicator of high sodium intake [16].
Ln 184 Is the formula located in the correct position?
Ln 276 & 279 “blow” below?
Ln 273-281 See general comments on factors influencing regional levels of gastric cancer associated with high salt intake.
Ln 337 “unites” units or unities?
Ln 341 Comment on any literature data on <25-year-olds.
Ln 344-351 “Our findings are consistent with previous studies [16]. It could possibly be explained by the reason that males excrete more salt through the urine (247 mmol/d) than do females (218 mmol/d) [24]. Moreover, females were considerably less susceptible to the negative effects of high sodium intake exposure since females are more likely to effectively maintain Na+ homeostasis during the acclimatization to the challenges of high salt dietary habits [25].”
This statement makes too many logical jumps. The higher sodium excretion by men than women indicated that men are ingesting less salt. Women will generally have lower food intake and, hence, lower exposure to salt. Clarify the statement concerning the effects of gender on susceptibility to gastric cancer.
Ln 359-361 “Nevertheless, in the few regions where a decline in sodium intake was observed during this period, this may be insufficient to overcome the increase in the cases number due to population growth and ageing.” Clarify and expand on this.
Ln 367-370 Speculate on the regional lifestyles and dietary practices that lead to differences in salt intake.
Ln 389 What lifestyle factors may influence or exacerbate the effects of salt on the incidence of gastric cancer?
J. Li, X.H. Kuang, Y. Zhang, D.M. Hu, K. Liu. Global burden of gastric cancer in adolescents and young adults: estimates from GLOBOCAN 2020. Public Health, Volume 210, 2022, Pages 58-64. https://doi.org/10.1016/j.puhe.2022.06.010.
Hagag S, Habib E, Tawfik S. Assessment of Knowledge and Practices toward Salt Intake among Adolescents. Open Access Maced J Med Sci [Internet]. 2022 Mar. 20 [cited 2023 Oct. 17];10(E):921-5. https://oamjms.eu/index.php/mjms/article/view/9081
Comments on the Quality of English Language
Minor changes are required. See general comments.
Author Response
For research article
Response to Reviewer X Comments
|
||
1. Summary |
|
|
Thank you very much for taking the time to review this manuscript. Please find the detailed responses below and the corresponding revisions/corrections highlighted/in track changes in the re-submitted files.
|
||
2. Questions for General Evaluation |
Reviewer’s Evaluation |
Response and Revisions |
Does the introduction provide sufficient background and include all relevant references? |
Can be improved |
We have made some some modifications. |
Are all the cited references relevant to the research? |
Yes |
|
Is the research design appropriate? |
Yes |
|
Are the methods adequately described? |
Yes |
|
Are the results clearly presented? |
Yes |
|
Are the conclusions supported by the results? |
Can be improved |
We have made some critical changes. |
3. Point-by-point response to Comments and Suggestions for Authors |
||
Comments 1: The study provides an excellent data reference for medical and health policymakers in identifying the overall scale of the ongoing problem in China and highlighting regions or demographics which need support. However, there is little or no discussion as to why salt intakes remain intransigently high or why there are significant regional differences. |
||
Response: Thank you for pointing this out. We agree with this comment. Therefore, we have added the discussion as to why salt intakes remain high or why there are significant regional differences.The content is as follows: ”This could be taken into account low SES there, a characteristic concerning a wide part of population. In particular, in low-income and middle-income regions sodium intake is more diffuse, diverse, and higher compared to those high-income regions[28]. The local residents in Shandong, one region close to The Yellow River Delta, are prone to have sea-foods and pickled foods with more sodium ingredients. Additionally, eating out has become popular than ever before and restaurant dishes often contain high salt. Also, the consumption of prepackaged food continues to increase and increases the exposure to high-sodium[32]. Whereas, the lowest disease burden were detected in Hong Kong and Macao across China. This may be related to the light diet for residents in Hong Kong and Macao. Those population consume less processed meat and pickled foods since the residents pay more attention to dietary patterns. Lifestyle modifications, alcohol drinking and high fat diet, could boost the effects of high sodium intake on gastric cancer and exert mediating effects. Diminishing the sodium intake would also scale back gastric cancer risk due to the emerging interaction effect. ”– page 9, line 409 to 423.
|
||
Comments 2: What proportion of the salt intake is considered added to food after the meal's preparation? What proportion is a constituent of a traditionally prepared meal? If the latter, the approaches needed to reduce intake will require intensive education and training to prepare low-salt alternatives. The incidences increase with ageing. So, do rural populations with an ageing population have exceptionally high salt intakes (present and lifetime) and high gastric cancer rates? |
||
Response: Thank you for pointing this out. We totally agree with your comments. However, our data does not include rural areas because of availability. As you suggested, we have added the description in the part of limitation. The content is as follows: “ And, rural regions could not be examined separately because of data availability as rural populations with an aging population have exceptionally high salt intakes across the lifetime[37]. ”– page 10, line 460 to 462.
|
||
Comments 3: The authors indicate no data on salt intake or gastric cancer for adolescents (<25 yr). See Li et al, 2022 and Hagag et al, 2022.Acquiring this information would be particularly important because the gut undergoes significant cellular and structural changes and is highly susceptible to dietary insults. High salt intake at this early stage of life may predispose the stomach to disruption and cancer in later life. Also, conditioning to salt in early life may affect taste bud sensitivity and the need/tolerance for salt in the diet in later life. Urbanization of the population is a significant change in society. Is there any indication that this is affecting the levels of salt intake or gastric cancer? |
||
Response: Agree. We have, accordingly, supplemented the effects of high salt in adolescence to emphasize this point. The content is as follows: ”High salt intake at this early stage of life may predispose the stomach to disruption and accelerate cancer development in later life[28]. Stomach is highly susceptible to dietary insults since the gut undergoes significant cellular and structural changes. Conditioning to salt in early life may affect taste bud sensitivity and influence the need and tolerance for salt in later life. It takes longer to develop gastric cancer for a sequence of preneoplastic lesions in older ages. A prolonged exposure to high sodium intake could potentially increase the susceptibility to gastric cancer among the elderly compared to the young population in the special time point. Population aging could also be a great influential factor on the burden of whole diseases, including the burden of gastric cancer attributable to high-salt diets[29]. China has implemented effective sodium reduction measures since 2007, such as accelerating the revision of the general rules of nutrition labeling of prepackaged foods and normalizing nutrition and health science education activities, and this may decrease the disease burden. Consequently, the average sodium daily intake in Chinese residents decreased from 5.9g in 2002 to 5.2g in 2012[30]. Although the salt intake of residents showed a downward trend, it was still 75% higher than the 6g/d recommended by the Chinese Dietary Guidelines (2016)[31]. Most importantly, the decreasing burden of gastric cancer attributable to high sodium intake may not offset the effects of more and more serious population. Therefore, the approaches needed to reduce intake will require intensive education and training to prepare low-salt alternatives. Individuals who have preference for salty foods need to have education regarding appropriate sodium intake and a low-sodium diet should be encouraged, especially among the seniors. ”– page 9, line 383 to 405.
As you suggested, we have added the reference in the part of reference. 28 Li J, Kuang XH, Zhang Y, Hu DM, Liu K: Global burden of gastric cancer in adolescents and young adults: estimates from GLOBOCAN 2020. Public Health 2022, 210:58-64.
|
||
Comments 4: “J:[email protected]) Is this meant to be here? |
||
Response: Thank you for pointing this out. We have, accordingly, deleted it.
|
||
Comments 5: “A healthy diet and lifestyle are required, especially taste preference and dietary behaviors.” Is this sentence necessary? |
||
Response: Thank you for pointing this out. The sentence has been deleted.
|
||
Comments 6: Accepting that high salt intake is the primary factor, add a paragraph on other factors that may contribute to or amplify the effects of sodium. |
||
Response: Thank for your valuable feedback that could improve the quality of our manuscript. We have, accordingly, added the description to make this point clear in the part of introduction. The content is as follows: ““High sodium intake accounts for many gastric cancer cases and also contribute to the disease burden. Previous evidence suggests that excess salt may promote gastric Helicobacter pylori (H. pylori) colonization in the stomach is a well-known risk factor for gastric cancer[8]. H. pylori infection constitutes an additional risk factor of gastric cancer with an obvious impact on the population with the highest daily sodium intake. A comprehensive mechanisms come into play by fostering the development of gastric adenocarcinoma owing to the synergy between an high sodium intake and H. pylori colonization. Collectively, the multitude mechanisms encompass the disruption of mucosal barriers, cellular integrity, modulation of H. pylori gene expression, oxidative stress induction, and provocation of inflammatory responses. ”– page 2, line 81 to 90. |
||
Comments 7: Ln 100 “s.”. Delete. |
||
Response: We have carefully checked the manuscript and deleted the word delete in the manuscript.
|
||
Comments 8: Ln 127-128 “High sodium intake was defined as an average 24-h urinary sodium excretion (grams per day) greater than 3g (95%UI:1g,5g) [16].”
An average 24-hour urinary sodium excretion (grams per day) of greater than 3g (95%UI:1g,5g) was used as a clear indicator of high sodium intake [16]. |
||
Response 8:Thank you for pointing this out. The reviewer is correct, and we have changed the revision as follows.”An average 24-hour urinary sodium excretion (grams per day) of greater than 3g (95%UI:1g,5g) was used as a clear indicator of high sodium intake [16].”– page 3, line 140.
|
||
Comments 9: Ln 184 Is the formula located in the correct position? |
||
Response 9: We have carefully checked the manuscript and corrected the errors accordingly. ””– page 4, line 166.
|
||
Comments 10: Ln 276 & 279 “blow” below? |
||
Response 10: We have carefully checked the manuscript and deleted the word in the manuscript.
|
||
Comments 11: Ln 273-281 See general comments on factors influencing regional levels of gastric cancer associated with high salt intake. |
||
Response:We have great thanks for your professional review on our article. According to your suggestions, we have made extensive corrections to our previous draft, and the detailed corrections are listed below.”This could be taken into account low SES there, a characteristic concerning a wide part of population. In particular, in low-income and middle-income regions sodium intake is more diffuse, diverse, and higher compared to those high-income regions[28]. The local residents in Shandong, one region close to The Yellow River Delta, are prone to have sea-foods and pickled foods with more sodium ingredients. Additionally, eating out has become popular than ever before and restaurant dishes often contain high salt. Also, the consumption of prepackaged food continues to increase and increases the exposure to high-sodium[32]. Whereas, the lowest disease burden were detected in Hong Kong and Macao across China. This may be related to the light diet for residents in Hong Kong and Macao. Those population consume less processed meat and pickled foods since the residents pay more attention to dietary patterns. Lifestyle modifications, alcohol drinking and high fat diet, could boost the effects of high sodium intake on gastric cancer and exert mediating effects. Diminishing the sodium intake would also scale back gastric cancer risk due to the emerging interaction effect. Dietary preference in daily life and demographic variables, including expanded life expectancy and aging, might be involved in serious disease burden of gastric cancer. ”– page 9, line 409-425. |
||
Comments 12: Ln 337 “unites” units or unities? |
||
Response: We have carefully checked the manuscript and corrected the errors accordingly. – page 8, line 358.
|
||
Comments 13: Ln 341 Comment on any literature data on <25-year-olds. |
||
Response: Agree. We have, accordingly, added description to emphasize this point. We sincerely appreciate the valuable comments. We have checked the literature carefully and added more references on literature data on <25-year-olds into the Discussion part in the revised manuscript.– page 9, line 412 and page 10, line 416. The added references are as follows: 28.Li J, Kuang XH, Zhang Y, Hu DM, Liu K: Global burden of gastric cancer in adolescents and young adults: estimates from GLOBOCAN 2020. Public Health 2022, 210:58-64. 32.Hagag S, Habib E, Tawfik S: Assessment of Knowledge and Practices toward Salt Intake among Adolescents. Open Access Macedonian Journal of Medical Sciences 2022, 10:921-925. |
||
Comments 14: Ln 344-351 “Our findings are consistent with previous studies [16]. It could possibly be explained by the reason that males excrete more salt through the urine (247 mmol/d) than do females (218 mmol/d) [24]. Moreover, females were considerably less susceptible to the negative effects of high sodium intake exposure since females are more likely to effectively maintain Na+ homeostasis during the acclimatization to the challenges of high salt dietary habits [25].” This statement makes too many logical jumps. The higher sodium excretion by men than women indicated that men are ingesting less salt. Women will generally have lower food intake and, hence, lower exposure to salt. Clarify the statement concerning the effects of gender on susceptibility to gastric cancer. |
||
Response: Agree. We have, accordingly, revised the description in the manuscript. The content is as follows: ”Our findings are consistent with previous studies[16]. Males could excrete more salt through the urine (247 mmol/d) than females(218 mmol/d)[24], indicating that males were likely to be exposed to higher sodium level than females as for internal environment and thus suffering from more serious disease burdens. Generally, males are prone to be involved in those negative lifestyle choices, such as smoking and alcohol abuse[25]. Smoking and alcohol consumption are well-known behavioral factors that are associated with unhealthy dietary patterns for gastric cancer risk. One previous study reported that smokers or alcohol drinkers prefer salty food. Smoking and alcohol consumption, as carcinogenic factors, could also exacerbate gastric cancer development when combining with unhealthy dietary habits. Especially, one of the mechanisms of alcohol consumption that affects gastric cancer is the production of inflammatory markers and the generation of oxidative stress substances, such as oxygen radical species[26]. ” We absolutely agree with you– page 9, line 364-376.
|
||
Comments 15: Ln 359-361 “Nevertheless, in the few regions where a decline in sodium intake was observed during this period, this may be insufficient to overcome the increase in the cases number due to population growth and ageing.” Clarify and expand on this. |
||
Response: As you suggested, we have made extensive modifications and added supplemented references to make the discussion convincing. ” High salt intake at this early stage of life may predispose the stomach to disruption and accelerate cancer development in later life[28]. Stomach is highly susceptible to dietary insults since the gut undergoes significant cellular and structural changes. Conditioning to salt in early life may affect taste bud sensitivity and influence the need and tolerance for salt in later life. It takes longer to develop gastric cancer for a sequence of preneoplastic lesions in older ages. A prolonged exposure to high sodium intake could potentially increase the susceptibility to gastric cancer among the elderly compared to the young population in the special time point. Population aging could also be a great influential factor on the burden of whole diseases, including the burden of gastric cancer attributable to high-salt diets[29]. China has implemented effective sodium reduction measures since 2007, such as accelerating the revision of the general rules of nutrition labeling of prepackaged foods and normalizing nutrition and health science education activities, and this may decrease the disease burden. Consequently, the average sodium daily intake in Chinese residents decreased from 5.9g in 2002 to 5.2g in 2012[30]. Although the salt intake of residents showed a downward trend, it was still 75% higher than the 6g/d recommended by the Chinese Dietary Guidelines (2016)[31]. Most importantly, the decreasing burden of gastric cancer attributable to high sodium intake may not offset the effects of more and more serious population. Therefore, the approaches needed to reduce intake will require intensive education and training to prepare low-salt alternatives. Individuals who have preference for salty foods need to have education regarding appropriate sodium intake and a low-sodium diet should be encouraged, especially among the seniors..”– page 9, line 383-405.
|
||
Comments 16: Ln 367-370 Speculate on the regional lifestyles and dietary practices that lead to differences in salt intake. |
||
Response: Agree. We have added description about regional lifestyles and dietary practices that lead to differences in salt intake to emphasize this point. “This could be taken into account low SES there, a characteristic concerning a wide part of population. In particular, in low-income and middle-income regions sodium intake is more diffuse, diverse, and higher compared to those high-income regions[28]. The local residents in Shandong, one region close to The Yellow River Delta, are prone to have sea-foods and pickled foods with more sodium ingredients. Additionally, eating out has become popular than ever before and restaurant dishes often contain high salt. Also, the consumption of prepackaged food continues to increase and increases the exposure to high-sodium[32]. Whereas, the lowest disease burden were detected in Hong Kong and Macao across China. This may be related to the light diet for residents in Hong Kong and Macao. Those population consume less processed meat and pickled foods since the residents pay more attention to dietary patterns. Lifestyle modifications, alcohol drinking and high fat diet, could boost the effects of high sodium intake on gastric cancer and exert mediating effects. Diminishing the sodium intake would also scale back gastric cancer risk due to the emerging interaction effect. ”– page 9, line 409 to 423.
|
||
Comments 17: Ln 389 What lifestyle factors may influence or exacerbate the effects of salt on the incidence of gastric cancer?
|
||
Response: Agree. We have, accordingly, discussed lifestyle to emphasize this point.”Additionally, eating out has become popular than ever before and restaurant dishes often contain high salt. Also, the consumption of prepackaged food continues to increase and increases the exposure to high-sodium[32]. Whereas, the lowest disease burden were detected in Hong Kong and Macao across China. This may be related to the light diet for residents in Hong Kong and Macao. Those population consume less processed meat and pickled foods since the residents pay more attention to dietary patterns. Lifestyle modifications, alcohol drinking and high fat diet, could boost the effects of high sodium intake on gastric cancer and exert mediating effects. Diminishing the sodium intake would also scale back gastric cancer risk due to the emerging interaction effect. Dietary preference in daily life and demographic variables, including expanded life expectancy and aging, might be involved in serious disease burden of gastric cancer. ”– page 9, line 414 to 425. |
||
4. Response to Comments on the Quality of English Language |
||
Point 1:Minor changes are required. See general comments. |
||
Response 1: Thanks for your suggestion. We have tried our best to polish the language in the revised manuscript. |
||
5. Additional clarifications |
||
|
||
|
Supplementary Tables 1. Deaths and ASMR of gastric cancer disease attributable-high sodium intake in 1990 and 2019 and the temporal trends from 1990-2019.
Characteristics |
1990 |
2019 |
1990-2019 |
|||
|
Deaths cases, No. (95% UI) |
ASMR per 100,000 No.(95% UI) |
Deaths cases, No. (95% UI) |
ASMR per 100,000 No. (95% UI) |
PAFs % (95% UI) |
EAPC(%) in ASMR No. (95% CI) |
Region |
||||||
Anhui |
2146.88 (46.72 to 7905.63) |
5.49 (0.12 to 20.37) |
2620.63 (55.77 to 9898) |
2.87 (0.06 to 10.85) |
8.89 (0.21 to 33.04) |
-2.03 (-2.44 to -1.63) |
Beijing |
174.73 (3.86 to 651.83) |
1.98 (0.04 to 7.31) |
252.11 (5.41 to 921.33) |
0.78 (0.02 to 2.85) |
8.90 (0.21 to 32.91) |
-3.32 (-3.41 to -3.23) |
Chongqing |
278.69 (6.04 to 1032.87) |
2.36 (0.05 to 8.71) |
643.60 (13.83 to 2461.93) |
1.52 (0.03 to 5.85) |
8.69 (0.21 to 32.50) |
-1.14 (-1.54 to -0.74) |
Fujian |
1042.99 (22.75 to 3940.85) |
5.22 (0.12 to 19.57) |
1234.75 (26.77 to 4643.99) |
2.47 (0.05 to 9.31) |
8.81(0.21 to 32.80) |
-2.63 (-2.72 to -2.54) |
Gansu |
612.58 (14.23 to 2391.08) |
4.46 (0.11 to 17.82) |
975.81 (22.56 to 3759.51) |
2.84 (0.07 to 10.99) |
8.04 (0.21 to 31.41) |
-1.44 (-1.89 to -0.98) |
Guangdong |
890.53 (19.77 to 3437.91) |
1.98 (0.04 to 7.65) |
1215.51 (27.48 to 4638.79) |
0.93 (0.02 to 3.55) |
8.59 (0.21 to 32.40) |
-2.23 (-2.68 to -1.77) |
Guangxi |
621.23 (13.68 to 2397.13) |
2.17 (0.05 to 8.51) |
1003.72 (22.01 to 3781.63) |
1.63 (0.04 to 6.17) |
8.46 (0.21 to 32.10) |
-0.38 (-0.83 to 0.07) |
Guizhou |
413.55 (9.05 to 1525.41) |
2.03 (0.05 to 7.56) |
672.22 (14.67 to 2516.46) |
1.57 (0.03 to 5.96) |
8.68 (0.21 to 32.59) |
-0.38 (-0.71 to -0.05) |
Hainan |
160.62 (3.58 to 615.73) |
3.60 (0.08 to 13.95) |
256.92 (5.75 to 989.52) |
2.30 (0.05 to 8.89) |
8.42 (0.21 to 32.14) |
-1.12 (-1.42 to -0.82) |
Hebei |
1509.07 (33.34 to 5589.76) |
3.41 (0.08 to 12.69) |
2080.59 (44.65 to 7700.39) |
2.04 (0.04 to 7.56) |
8.83 (0.21 to 32.91) |
-1.63 (-2.09 to -1.18) |
Heilongjiang |
589.98 (13.10 to 2252.24) |
2.98 (0.07 to 11.43) |
929.05 (20.52 to 3481.43) |
1.57 (0.04 to 6.02) |
8.52 (0.21 to 32.26) |
-1.76 (-2.05 to -1.46) |
Henan |
2611.86 (57.8 to 9561.77) |
4.28 (0.10 to 15.83) |
2950.71 (64.73 to 10986.83) |
2.41 (0.05 to 9.02) |
8.79 (0.21 to 32.91) |
-1.91 (-2.30 to -1.52) |
Hong Kong |
81.96 (1.92 to 299.89) |
1.49 (0.03 to 5.46) |
107.24 (2.29 to 409.98) |
0.72 (0.02 to 2.72) |
8.79 (0.21 to 32.88) |
-2.46 (-2.69 to -2.22) |
Hubei |
1035.70 (23.33 to 3810.60) |
2.77 (0.06 to 10.18) |
1492.15 (32.15 to 5669.98) |
1.73 (0.04 to 6.54) |
8.72 (0.21 to 32.71) |
-1.06 (-1.44 to -0.68) |
Hunan |
762.78 (17.22 to 2820.83) |
1.75 (0.04 to 6.47) |
1132.43 (24.55 to 4254.50) |
1.16 (0.03 to 4.36) |
8.75 (0.21 to 32.69) |
-0.96 (-1.25 to -0.66) |
Inner Mongolia |
357.90 (7.78 to 1336.06) |
2.86 (0.06 to 10.7) |
574.57 (12.24 to 2104.58) |
1.69 (0.04 to 6.19) |
8.88 (0.21 to 32.98) |
-1.73 (-2.08 to -1.38) |
Jiangsu |
2330.12 (52.07 to 8597.48) |
4.34 (0.10 to 16.1) |
3034.31 (65.72 to 11195.55) |
2.22 (0.05 to 8.17) |
8.82 (0.21 to 32.89) |
-2.14 (-2.65 to -1.62) |
Jiangxi |
846.71 (18.73 to 3142.73) |
3.45 (0.08 to 12.83) |
1016.95 (21.67 to 3815.86) |
1.80 (0.04 to 6.80) |
8.89 (0.21 to 33.01) |
-2.01 (-2.25 to -1.77) |
Jilin |
468.81 (10.31 to 1808.05) |
3.02 (0.07 to 11.66) |
530.16 (11.84 to 1939.54) |
1.25 (0.03 to 4.65) |
8.75 (0.21 to 32.75) |
-3.19 (-3.62 to -2.75) |
Liaoning |
859.80 (19.15 to 3286.72) |
3.08 (0.07 to 11.86) |
1355.75 (29.84 to 5145.03) |
1.76 (0.04 to 6.73) |
8.59 (0.21 to 32.39) |
-1.55 (-1.8 to -1.31) |
Macao |
4.15 (0.09 to 15.17) |
1.52 (0.03 to 5.52) |
5.83 (0.12 to 22.49) |
0.63 (0.01 to 2.42) |
8.83 (0.21 to 32.97) |
-3.14 (-3.34 to -2.95) |
Ningxia |
113.15 (2.47 to 438.24) |
4.82 (0.11 to 18.84) |
243.82 (5.48 to 937.69) |
3.41 (0.08 to 13.20) |
8.44 (0.21 to 32.27) |
-0.98 (-1.13 to -0.82) |
Qinghai |
127.81 (2.69 to 476.30) |
5.44 (0.12 to 20.04) |
273.51 (5.83 to 1019.98) |
4.36 (0.10 to 16.29) |
8.88 (0.21 to 32.97) |
-0.51 (-0.71 to -0.3) |
Shaanxi |
796.28 (17.31 to 2954.11) |
3.61 (0.08 to 13.51) |
1044.35 (22.34 to 3865.56) |
1.98 (0.04 to 7.39) |
8.84 (0.21 to 32.83) |
-1.94 (-2.31 to -1.58) |
Shandong |
2363.43 (51.86 to 8795.77) |
3.73 (0.08 to 14.01) |
3543.69 (78.12 to 13024.72) |
2.31 (0.05 to 8.52) |
8.86 (0.21 to 32.97) |
-1.35 (-1.88 to -0.82) |
Shanghai |
492.02 (10.58 to 1852.30) |
3.50 (0.08 to 13.27) |
566.30 (12.04 to 2139.24) |
1.34 (0.03 to 5.06) |
8.70 (0.21 to 32.66) |
-3.34 (-3.43 to -3.25) |
Shanxi |
926.59 (20.74 to 3533.39) |
4.64 (0.11 to 17.87) |
1494.43 (32.86 to 5639.88) |
3.16 (0.07 to 12.10) |
8.59 (0.21 to 32.38) |
-1.05 (-1.52 to -0.57) |
Sichuan |
2491.59 (54.53 to 9295.86) |
3.11 (0.07 to 11.65) |
2747.61 (59.9 to 10112.62) |
2.09 (0.05 to 7.76) |
8.70 (0.21 to 32.57) |
-1.14(-1.91 to -0.36) |
Tianjin |
144.33 (3.25 to 539.00) |
2.04 (0.05 to 7.69) |
199.11 (4.28 to 773.61) |
0.94 (0.02 to 3.63) |
8.60 (0.21 to 32.48) |
-2.49 (-2.67 to -2.32) |
Tibet |
60.52 (1.32 to 228.81) |
4.12 (0.09 to 15.49) |
62.28 (1.31 to 227.56) |
2.35 (0.05 to 8.60) |
8.92 (0.21 to 32.95) |
-2.13 (-2.37 to -1.89) |
Xinjiang |
269.27 (5.72 to 1019.43) |
3.19 (0.07 to 12.12) |
559.31 (11.97 to 2077.37) |
2.34 (0.05 to 8.76) |
8.88 (0.21 to 32.84) |
-0.75 (-0.96 to -0.54) |
Yunnan |
385.05 (8.44 to 1452.29) |
1.59 (0.04 to 6.00) |
677.78 (14.8 to 2498.95) |
1.24 (0.03 to 4.61) |
8.71 (0.21 to 32.70) |
-0.36 (-0.69 to -0.02) |
Zhejiang |
1255.84 (27 to 4654.13) |
3.68 (0.08 to 13.67) |
1634.26 (34.75 to 5953.92) |
1.85 (0.04 to 6.76) |
9.00 (0.21 to 33.13) |
-2.14 (-2.44 to -1.83) |
ASMR=age-standardized mortality rate; PAF=population attributable fraction; EAPC=estimated annual percentage change.
Supplementary Tables 2. DALYs and ASDR of gastric cancer disease attributable to high sodium intake in 1990 and 2019 and the temporal trends from 1990 to 2019.
Characteristics |
1990 |
2019 |
1990-2019 |
|||
|
Deaths cases, No. (95% UI) |
ASDR per 100,000 No.(95% UI) |
DALYs, No. (95% UI) |
ASDR per 100,000 No. (95% UI) |
PAFs % (95% UI) |
EAPC(%) in ASMR No. (95% CI) |
Region |
||||||
Anhui |
2146.88 (46.72 to 7905.63) |
5.49 (0.12 to 20.37) |
2620.63 (55.77 to 9898) |
2.87 (0.06 to 10.85) |
8.89 (0.21 to 33.04) |
-2.03 (-2.44 to -1.63) |
Beijing |
174.73 (3.86 to 651.83) |
1.98 (0.04 to 7.31) |
252.11 (5.41 to 921.33) |
0.78 (0.02 to 2.85) |
8.90 (0.21 to 32.91) |
-3.32 (-3.41 to -3.23) |
Chongqing |
278.69 (6.04 to 1032.87) |
2.36 (0.05 to 8.71) |
643.60 (13.83 to 2461.93) |
1.52 (0.03 to 5.85) |
8.69 (0.21 to 32.50) |
-1.14 (-1.54 to -0.74) |
Fujian |
1042.99 (22.75 to 3940.85) |
5.22 (0.12 to 19.57) |
1234.75 (26.77 to 4643.99) |
2.47 (0.05 to 9.31) |
8.81(0.21 to 32.80) |
-2.63 (-2.72 to -2.54) |
Gansu |
612.58 (14.23 to 2391.08) |
4.46 (0.11 to 17.82) |
975.81 (22.56 to 3759.51) |
2.84 (0.07 to 10.99) |
8.04 (0.21 to 31.41) |
-1.44 (-1.89 to -0.98) |
Guangdong |
890.53 (19.77 to 3437.91) |
1.98 (0.04 to 7.65) |
1215.51 (27.48 to 4638.79) |
0.93 (0.02 to 3.55) |
8.59 (0.21 to 32.40) |
-2.23 (-2.68 to -1.77) |
Guangxi |
621.23 (13.68 to 2397.13) |
2.17 (0.05 to 8.51) |
1003.72 (22.01 to 3781.63) |
1.63 (0.04 to 6.17) |
8.46 (0.21 to 32.10) |
-0.38 (-0.83 to 0.07) |
Guizhou |
413.55 (9.05 to 1525.41) |
2.03 (0.05 to 7.56) |
672.22 (14.67 to 2516.46) |
1.57 (0.03 to 5.96) |
8.68 (0.21 to 32.59) |
-0.38 (-0.71 to -0.05) |
Hainan |
160.62 (3.58 to 615.73) |
3.60 (0.08 to 13.95) |
256.92 (5.75 to 989.52) |
2.30 (0.05 to 8.89) |
8.42 (0.21 to 32.14) |
-1.12 (-1.42 to -0.82) |
Hebei |
1509.07 (33.34 to 5589.76) |
3.41 (0.08 to 12.69) |
2080.59 (44.65 to 7700.39) |
2.04 (0.04 to 7.56) |
8.83 (0.21 to 32.91) |
-1.63 (-2.09 to -1.18) |
Heilongjiang |
589.98 (13.10 to 2252.24) |
2.98 (0.07 to 11.43) |
929.05 (20.52 to 3481.43) |
1.57 (0.04 to 6.02) |
8.52 (0.21 to 32.26) |
-1.76 (-2.05 to -1.46) |
Henan |
2611.86 (57.8 to 9561.77) |
4.28 (0.10 to 15.83) |
2950.71 (64.73 to 10986.83) |
2.41 (0.05 to 9.02) |
8.79 (0.21 to 32.91) |
-1.91 (-2.30 to -1.52) |
Hong Kong |
81.96 (1.92 to 299.89) |
1.49 (0.03 to 5.46) |
107.24 (2.29 to 409.98) |
0.72 (0.02 to 2.72) |
8.79 (0.21 to 32.88) |
-2.46 (-2.69 to -2.22) |
Hubei |
1035.70 (23.33 to 3810.60) |
2.77 (0.06 to 10.18) |
1492.15 (32.15 to 5669.98) |
1.73 (0.04 to 6.54) |
8.72 (0.21 to 32.71) |
-1.06 (-1.44 to -0.68) |
Hunan |
762.78 (17.22 to 2820.83) |
1.75 (0.04 to 6.47) |
1132.43 (24.55 to 4254.50) |
1.16 (0.03 to 4.36) |
8.75 (0.21 to 32.69) |
-0.96 (-1.25 to -0.66) |
Inner Mongolia |
357.90 (7.78 to 1336.06) |
2.86 (0.06 to 10.7) |
574.57 (12.24 to 2104.58) |
1.69 (0.04 to 6.19) |
8.88 (0.21 to 32.98) |
-1.73 (-2.08 to -1.38) |
Jiangsu |
2330.12 (52.07 to 8597.48) |
4.34 (0.10 to 16.1) |
3034.31 (65.72 to 11195.55) |
2.22 (0.05 to 8.17) |
8.82 (0.21 to 32.89) |
-2.14 (-2.65 to -1.62) |
Jiangxi |
846.71 (18.73 to 3142.73) |
3.45 (0.08 to 12.83) |
1016.95 (21.67 to 3815.86) |
1.80 (0.04 to 6.80) |
8.89 (0.21 to 33.01) |
-2.01 (-2.25 to -1.77) |
Jilin |
468.81 (10.31 to 1808.05) |
3.02 (0.07 to 11.66) |
530.16 (11.84 to 1939.54) |
1.25 (0.03 to 4.65) |
8.75 (0.21 to 32.75) |
-3.19 (-3.62 to -2.75) |
Liaoning |
859.80 (19.15 to 3286.72) |
3.08 (0.07 to 11.86) |
1355.75 (29.84 to 5145.03) |
1.76 (0.04 to 6.73) |
8.59 (0.21 to 32.39) |
-1.55 (-1.8 to -1.31) |
Macao |
4.15 (0.09 to 15.17) |
1.52 (0.03 to 5.52) |
5.83 (0.12 to 22.49) |
0.63 (0.01 to 2.42) |
8.83 (0.21 to 32.97) |
-3.14 (-3.34 to -2.95) |
Ningxia |
113.15 (2.47 to 438.24) |
4.82 (0.11 to 18.84) |
243.82 (5.48 to 937.69) |
3.41 (0.08 to 13.20) |
8.44 (0.21 to 32.27) |
-0.98 (-1.13 to -0.82) |
Qinghai |
127.81 (2.69 to 476.30) |
5.44 (0.12 to 20.04) |
273.51 (5.83 to 1019.98) |
4.36 (0.10 to 16.29) |
8.88 (0.21 to 32.97) |
-0.51 (-0.71 to -0.3) |
Shaanxi |
796.28 (17.31 to 2954.11) |
3.61 (0.08 to 13.51) |
1044.35 (22.34 to 3865.56) |
1.98 (0.04 to 7.39) |
8.84 (0.21 to 32.83) |
-1.94 (-2.31 to -1.58) |
Shandong |
2363.43 (51.86 to 8795.77) |
3.73 (0.08 to 14.01) |
3543.69 (78.12 to 13024.72) |
2.31 (0.05 to 8.52) |
8.86 (0.21 to 32.97) |
-1.35 (-1.88 to -0.82) |
Shanghai |
492.02 (10.58 to 1852.30) |
3.50 (0.08 to 13.27) |
566.30 (12.04 to 2139.24) |
1.34 (0.03 to 5.06) |
8.70 (0.21 to 32.66) |
-3.34 (-3.43 to -3.25) |
Shanxi |
926.59 (20.74 to 3533.39) |
4.64 (0.11 to 17.87) |
1494.43 (32.86 to 5639.88) |
3.16 (0.07 to 12.10) |
8.59 (0.21 to 32.38) |
-1.05 (-1.52 to -0.57) |
Sichuan |
2491.59 (54.53 to 9295.86) |
3.11 (0.07 to 11.65) |
2747.61 (59.9 to 10112.62) |
2.09 (0.05 to 7.76) |
8.70 (0.21 to 32.57) |
-1.14(-1.91 to -0.36) |
Tianjin |
144.33 (3.25 to 539.00) |
2.04 (0.05 to 7.69) |
199.11 (4.28 to 773.61) |
0.94 (0.02 to 3.63) |
8.60 (0.21 to 32.48) |
-2.49 (-2.67 to -2.32) |
Tibet |
60.52 (1.32 to 228.81) |
4.12 (0.09 to 15.49) |
62.28 (1.31 to 227.56) |
2.35 (0.05 to 8.60) |
8.92 (0.21 to 32.95) |
-2.13 (-2.37 to -1.89) |
Xinjiang |
269.27 (5.72 to 1019.43) |
3.19 (0.07 to 12.12) |
559.31 (11.97 to 2077.37) |
2.34 (0.05 to 8.76) |
8.88 (0.21 to 32.84) |
-0.75 (-0.96 to -0.54) |
Yunnan |
385.05 (8.44 to 1452.29) |
1.59 (0.04 to 6.00) |
677.78 (14.8 to 2498.95) |
1.24 (0.03 to 4.61) |
8.71 (0.21 to 32.70) |
-0.36 (-0.69 to -0.02) |
Zhejiang |
1255.84 (27 to 4654.13) |
3.68 (0.08 to 13.67) |
1634.26 (34.75 to 5953.92) |
1.85 (0.04 to 6.76) |
9.00 (0.21 to 33.13) |
-2.14 (-2.44 to -1.83) |
DALYs=disability-adjusted life years; ASDR=age-standardized DALYs rate; PAF=population attributable fraction; EAPC=estimated annual percentage change.

Reviewer 3 Report
Comments and Suggestions for Authors
The manuscript shows the landscape of death, DALY, ASMR and ASDR in high sodium intake-related gastric cancer in China and provides a good foundation and overlook for gastric cancer research and therapy. Here is my concerns which will be helpful to improve this study.
In Fig.3A and 3B, both of the DALY rate and the death rate have some fluctuation in different SDI regions. However, they lacks some statistic significance and authors should provide some statistic values for the quantitative data.
In fig 4, authors showed that the ASMR and ASDR declined with the increment of SDI. However, different regions have respective individual SDI and one plot included all regions confuses the reader. Authors should show the connections based on high, middle and low SDI seperately to identify that the trends in different SDI regions.
Comments on the Quality of English Language
The writing has some mistakes and the quality of language can be improved.
Author Response
For research article
Response to Reviewer X Comments
|
||
1. Summary |
|
|
Thank you very much for taking the time to review this manuscript. Please find the detailed responses below and the corresponding revisions/corrections highlighted/in track changes in the re-submitted files.
|
||
2. Questions for General Evaluation |
Reviewer’s Evaluation |
Response and Revisions |
Does the introduction provide sufficient background and include all relevant references? |
Yes |
|
Are all the cited references relevant to the research? |
Can be improved |
Thank you for pointing this out. We have made some changes. |
Is the research design appropriate? |
|
|
Are the methods adequately described? |
|
|
Are the results clearly presented? |
Yes |
|
Are the conclusions supported by the results? |
|
|
3. Point-by-point response to Comments and Suggestions for Authors |
||
Comments 1: In Fig.3A and 3B, both of the DALY rate and the death rate have some fluctuation in different SDI regions. However, they lacks some statistic significance and authors should provide some statistic values for the quantitative data.
|
||
Response 1: Thank you for pointing this out. We agree with this comment. Therefore, we have added a description of tabular statistics.”Generally, DALYs rate in national level and global level were relatively stable, and EAPCs in DALYs rate were slightly decreasing from 1990 to 2019 in all regions across China(Ptrend>0.05)(Fig.3A). Specifically, in the high SDI regions, the trend of age-specific DALYs rate was downward in individuals ≤90 and upward in individuals >90 from 1990 to 2019, among which the biggest increasement and drop occurred in those individuals aged 40-44 and 90-94, respectively; in the low-middle SDI regions, the trend of age-specific DALYs rate was downward in individuals ≤80 and upward in individuals aged 80 over with biggest increasement and decrement in individuals aged 85-90 and 45-50. A similar trend was detected for EAPC in the mortality rate (Fig.3B). ”– page 5, line 292 to 300. |
||
Comments 2: In fig 4, authors showed that the ASMR and ASDR declined with the increment of SDI. However, different regions have respective individual SDI and one plot included all regions confuses the reader. Authors should show the connections based on high, middle and low SDI seperately to identify that the trends in different SDI regions. |
||
Response 2: Thank you for pointing this out. It could be more clear to show the connections based on high, middle and low SDI separately to identify that the trends in different SDI regions. However, identifying the trends in different regions are really more specific. The figure 4 provide he trends in different regions, but not the details of the ASMR and ASDR. Thank you for your suggestions. |
||
4. Response to Comments on the Quality of English Language |
||
Point 1:The writing has some mistakes and the quality of language can be improved. |
||
Response 1: We tried our best to improve the quality of language in the manuscript and polish it. These changes could not influence the content and framework of the paper. And, we marked the revised content in red in the revised manuscript. We really appreciate for your wonderful work. |
||
5. Additional clarifications |
||
|
Supplementary Tables 1. Deaths and ASMR of gastric cancer disease attributable-high sodium intake in 1990 and 2019 and the temporal trends from 1990-2019.
Characteristics |
1990 |
2019 |
1990-2019 |
|||
|
Deaths cases, No. (95% UI) |
ASMR per 100,000 No.(95% UI) |
Deaths cases, No. (95% UI) |
ASMR per 100,000 No. (95% UI) |
PAFs % (95% UI) |
EAPC(%) in ASMR No. (95% CI) |
Region |
||||||
Anhui |
2146.88 (46.72 to 7905.63) |
5.49 (0.12 to 20.37) |
2620.63 (55.77 to 9898) |
2.87 (0.06 to 10.85) |
8.89 (0.21 to 33.04) |
-2.03 (-2.44 to -1.63) |
Beijing |
174.73 (3.86 to 651.83) |
1.98 (0.04 to 7.31) |
252.11 (5.41 to 921.33) |
0.78 (0.02 to 2.85) |
8.90 (0.21 to 32.91) |
-3.32 (-3.41 to -3.23) |
Chongqing |
278.69 (6.04 to 1032.87) |
2.36 (0.05 to 8.71) |
643.60 (13.83 to 2461.93) |
1.52 (0.03 to 5.85) |
8.69 (0.21 to 32.50) |
-1.14 (-1.54 to -0.74) |
Fujian |
1042.99 (22.75 to 3940.85) |
5.22 (0.12 to 19.57) |
1234.75 (26.77 to 4643.99) |
2.47 (0.05 to 9.31) |
8.81(0.21 to 32.80) |
-2.63 (-2.72 to -2.54) |
Gansu |
612.58 (14.23 to 2391.08) |
4.46 (0.11 to 17.82) |
975.81 (22.56 to 3759.51) |
2.84 (0.07 to 10.99) |
8.04 (0.21 to 31.41) |
-1.44 (-1.89 to -0.98) |
Guangdong |
890.53 (19.77 to 3437.91) |
1.98 (0.04 to 7.65) |
1215.51 (27.48 to 4638.79) |
0.93 (0.02 to 3.55) |
8.59 (0.21 to 32.40) |
-2.23 (-2.68 to -1.77) |
Guangxi |
621.23 (13.68 to 2397.13) |
2.17 (0.05 to 8.51) |
1003.72 (22.01 to 3781.63) |
1.63 (0.04 to 6.17) |
8.46 (0.21 to 32.10) |
-0.38 (-0.83 to 0.07) |
Guizhou |
413.55 (9.05 to 1525.41) |
2.03 (0.05 to 7.56) |
672.22 (14.67 to 2516.46) |
1.57 (0.03 to 5.96) |
8.68 (0.21 to 32.59) |
-0.38 (-0.71 to -0.05) |
Hainan |
160.62 (3.58 to 615.73) |
3.60 (0.08 to 13.95) |
256.92 (5.75 to 989.52) |
2.30 (0.05 to 8.89) |
8.42 (0.21 to 32.14) |
-1.12 (-1.42 to -0.82) |
Hebei |
1509.07 (33.34 to 5589.76) |
3.41 (0.08 to 12.69) |
2080.59 (44.65 to 7700.39) |
2.04 (0.04 to 7.56) |
8.83 (0.21 to 32.91) |
-1.63 (-2.09 to -1.18) |
Heilongjiang |
589.98 (13.10 to 2252.24) |
2.98 (0.07 to 11.43) |
929.05 (20.52 to 3481.43) |
1.57 (0.04 to 6.02) |
8.52 (0.21 to 32.26) |
-1.76 (-2.05 to -1.46) |
Henan |
2611.86 (57.8 to 9561.77) |
4.28 (0.10 to 15.83) |
2950.71 (64.73 to 10986.83) |
2.41 (0.05 to 9.02) |
8.79 (0.21 to 32.91) |
-1.91 (-2.30 to -1.52) |
Hong Kong |
81.96 (1.92 to 299.89) |
1.49 (0.03 to 5.46) |
107.24 (2.29 to 409.98) |
0.72 (0.02 to 2.72) |
8.79 (0.21 to 32.88) |
-2.46 (-2.69 to -2.22) |
Hubei |
1035.70 (23.33 to 3810.60) |
2.77 (0.06 to 10.18) |
1492.15 (32.15 to 5669.98) |
1.73 (0.04 to 6.54) |
8.72 (0.21 to 32.71) |
-1.06 (-1.44 to -0.68) |
Hunan |
762.78 (17.22 to 2820.83) |
1.75 (0.04 to 6.47) |
1132.43 (24.55 to 4254.50) |
1.16 (0.03 to 4.36) |
8.75 (0.21 to 32.69) |
-0.96 (-1.25 to -0.66) |
Inner Mongolia |
357.90 (7.78 to 1336.06) |
2.86 (0.06 to 10.7) |
574.57 (12.24 to 2104.58) |
1.69 (0.04 to 6.19) |
8.88 (0.21 to 32.98) |
-1.73 (-2.08 to -1.38) |
Jiangsu |
2330.12 (52.07 to 8597.48) |
4.34 (0.10 to 16.1) |
3034.31 (65.72 to 11195.55) |
2.22 (0.05 to 8.17) |
8.82 (0.21 to 32.89) |
-2.14 (-2.65 to -1.62) |
Jiangxi |
846.71 (18.73 to 3142.73) |
3.45 (0.08 to 12.83) |
1016.95 (21.67 to 3815.86) |
1.80 (0.04 to 6.80) |
8.89 (0.21 to 33.01) |
-2.01 (-2.25 to -1.77) |
Jilin |
468.81 (10.31 to 1808.05) |
3.02 (0.07 to 11.66) |
530.16 (11.84 to 1939.54) |
1.25 (0.03 to 4.65) |
8.75 (0.21 to 32.75) |
-3.19 (-3.62 to -2.75) |
Liaoning |
859.80 (19.15 to 3286.72) |
3.08 (0.07 to 11.86) |
1355.75 (29.84 to 5145.03) |
1.76 (0.04 to 6.73) |
8.59 (0.21 to 32.39) |
-1.55 (-1.8 to -1.31) |
Macao |
4.15 (0.09 to 15.17) |
1.52 (0.03 to 5.52) |
5.83 (0.12 to 22.49) |
0.63 (0.01 to 2.42) |
8.83 (0.21 to 32.97) |
-3.14 (-3.34 to -2.95) |
Ningxia |
113.15 (2.47 to 438.24) |
4.82 (0.11 to 18.84) |
243.82 (5.48 to 937.69) |
3.41 (0.08 to 13.20) |
8.44 (0.21 to 32.27) |
-0.98 (-1.13 to -0.82) |
Qinghai |
127.81 (2.69 to 476.30) |
5.44 (0.12 to 20.04) |
273.51 (5.83 to 1019.98) |
4.36 (0.10 to 16.29) |
8.88 (0.21 to 32.97) |
-0.51 (-0.71 to -0.3) |
Shaanxi |
796.28 (17.31 to 2954.11) |
3.61 (0.08 to 13.51) |
1044.35 (22.34 to 3865.56) |
1.98 (0.04 to 7.39) |
8.84 (0.21 to 32.83) |
-1.94 (-2.31 to -1.58) |
Shandong |
2363.43 (51.86 to 8795.77) |
3.73 (0.08 to 14.01) |
3543.69 (78.12 to 13024.72) |
2.31 (0.05 to 8.52) |
8.86 (0.21 to 32.97) |
-1.35 (-1.88 to -0.82) |
Shanghai |
492.02 (10.58 to 1852.30) |
3.50 (0.08 to 13.27) |
566.30 (12.04 to 2139.24) |
1.34 (0.03 to 5.06) |
8.70 (0.21 to 32.66) |
-3.34 (-3.43 to -3.25) |
Shanxi |
926.59 (20.74 to 3533.39) |
4.64 (0.11 to 17.87) |
1494.43 (32.86 to 5639.88) |
3.16 (0.07 to 12.10) |
8.59 (0.21 to 32.38) |
-1.05 (-1.52 to -0.57) |
Sichuan |
2491.59 (54.53 to 9295.86) |
3.11 (0.07 to 11.65) |
2747.61 (59.9 to 10112.62) |
2.09 (0.05 to 7.76) |
8.70 (0.21 to 32.57) |
-1.14(-1.91 to -0.36) |
Tianjin |
144.33 (3.25 to 539.00) |
2.04 (0.05 to 7.69) |
199.11 (4.28 to 773.61) |
0.94 (0.02 to 3.63) |
8.60 (0.21 to 32.48) |
-2.49 (-2.67 to -2.32) |
Tibet |
60.52 (1.32 to 228.81) |
4.12 (0.09 to 15.49) |
62.28 (1.31 to 227.56) |
2.35 (0.05 to 8.60) |
8.92 (0.21 to 32.95) |
-2.13 (-2.37 to -1.89) |
Xinjiang |
269.27 (5.72 to 1019.43) |
3.19 (0.07 to 12.12) |
559.31 (11.97 to 2077.37) |
2.34 (0.05 to 8.76) |
8.88 (0.21 to 32.84) |
-0.75 (-0.96 to -0.54) |
Yunnan |
385.05 (8.44 to 1452.29) |
1.59 (0.04 to 6.00) |
677.78 (14.8 to 2498.95) |
1.24 (0.03 to 4.61) |
8.71 (0.21 to 32.70) |
-0.36 (-0.69 to -0.02) |
Zhejiang |
1255.84 (27 to 4654.13) |
3.68 (0.08 to 13.67) |
1634.26 (34.75 to 5953.92) |
1.85 (0.04 to 6.76) |
9.00 (0.21 to 33.13) |
-2.14 (-2.44 to -1.83) |
ASMR=age-standardized mortality rate; PAF=population attributable fraction; EAPC=estimated annual percentage change.
Supplementary Tables 2. DALYs and ASDR of gastric cancer disease attributable to high sodium intake in 1990 and 2019 and the temporal trends from 1990 to 2019.
Characteristics |
1990 |
2019 |
1990-2019 |
|||
|
Deaths cases, No. (95% UI) |
ASDR per 100,000 No.(95% UI) |
DALYs, No. (95% UI) |
ASDR per 100,000 No. (95% UI) |
PAFs % (95% UI) |
EAPC(%) in ASMR No. (95% CI) |
Region |
||||||
Anhui |
2146.88 (46.72 to 7905.63) |
5.49 (0.12 to 20.37) |
2620.63 (55.77 to 9898) |
2.87 (0.06 to 10.85) |
8.89 (0.21 to 33.04) |
-2.03 (-2.44 to -1.63) |
Beijing |
174.73 (3.86 to 651.83) |
1.98 (0.04 to 7.31) |
252.11 (5.41 to 921.33) |
0.78 (0.02 to 2.85) |
8.90 (0.21 to 32.91) |
-3.32 (-3.41 to -3.23) |
Chongqing |
278.69 (6.04 to 1032.87) |
2.36 (0.05 to 8.71) |
643.60 (13.83 to 2461.93) |
1.52 (0.03 to 5.85) |
8.69 (0.21 to 32.50) |
-1.14 (-1.54 to -0.74) |
Fujian |
1042.99 (22.75 to 3940.85) |
5.22 (0.12 to 19.57) |
1234.75 (26.77 to 4643.99) |
2.47 (0.05 to 9.31) |
8.81(0.21 to 32.80) |
-2.63 (-2.72 to -2.54) |
Gansu |
612.58 (14.23 to 2391.08) |
4.46 (0.11 to 17.82) |
975.81 (22.56 to 3759.51) |
2.84 (0.07 to 10.99) |
8.04 (0.21 to 31.41) |
-1.44 (-1.89 to -0.98) |
Guangdong |
890.53 (19.77 to 3437.91) |
1.98 (0.04 to 7.65) |
1215.51 (27.48 to 4638.79) |
0.93 (0.02 to 3.55) |
8.59 (0.21 to 32.40) |
-2.23 (-2.68 to -1.77) |
Guangxi |
621.23 (13.68 to 2397.13) |
2.17 (0.05 to 8.51) |
1003.72 (22.01 to 3781.63) |
1.63 (0.04 to 6.17) |
8.46 (0.21 to 32.10) |
-0.38 (-0.83 to 0.07) |
Guizhou |
413.55 (9.05 to 1525.41) |
2.03 (0.05 to 7.56) |
672.22 (14.67 to 2516.46) |
1.57 (0.03 to 5.96) |
8.68 (0.21 to 32.59) |
-0.38 (-0.71 to -0.05) |
Hainan |
160.62 (3.58 to 615.73) |
3.60 (0.08 to 13.95) |
256.92 (5.75 to 989.52) |
2.30 (0.05 to 8.89) |
8.42 (0.21 to 32.14) |
-1.12 (-1.42 to -0.82) |
Hebei |
1509.07 (33.34 to 5589.76) |
3.41 (0.08 to 12.69) |
2080.59 (44.65 to 7700.39) |
2.04 (0.04 to 7.56) |
8.83 (0.21 to 32.91) |
-1.63 (-2.09 to -1.18) |
Heilongjiang |
589.98 (13.10 to 2252.24) |
2.98 (0.07 to 11.43) |
929.05 (20.52 to 3481.43) |
1.57 (0.04 to 6.02) |
8.52 (0.21 to 32.26) |
-1.76 (-2.05 to -1.46) |
Henan |
2611.86 (57.8 to 9561.77) |
4.28 (0.10 to 15.83) |
2950.71 (64.73 to 10986.83) |
2.41 (0.05 to 9.02) |
8.79 (0.21 to 32.91) |
-1.91 (-2.30 to -1.52) |
Hong Kong |
81.96 (1.92 to 299.89) |
1.49 (0.03 to 5.46) |
107.24 (2.29 to 409.98) |
0.72 (0.02 to 2.72) |
8.79 (0.21 to 32.88) |
-2.46 (-2.69 to -2.22) |
Hubei |
1035.70 (23.33 to 3810.60) |
2.77 (0.06 to 10.18) |
1492.15 (32.15 to 5669.98) |
1.73 (0.04 to 6.54) |
8.72 (0.21 to 32.71) |
-1.06 (-1.44 to -0.68) |
Hunan |
762.78 (17.22 to 2820.83) |
1.75 (0.04 to 6.47) |
1132.43 (24.55 to 4254.50) |
1.16 (0.03 to 4.36) |
8.75 (0.21 to 32.69) |
-0.96 (-1.25 to -0.66) |
Inner Mongolia |
357.90 (7.78 to 1336.06) |
2.86 (0.06 to 10.7) |
574.57 (12.24 to 2104.58) |
1.69 (0.04 to 6.19) |
8.88 (0.21 to 32.98) |
-1.73 (-2.08 to -1.38) |
Jiangsu |
2330.12 (52.07 to 8597.48) |
4.34 (0.10 to 16.1) |
3034.31 (65.72 to 11195.55) |
2.22 (0.05 to 8.17) |
8.82 (0.21 to 32.89) |
-2.14 (-2.65 to -1.62) |
Jiangxi |
846.71 (18.73 to 3142.73) |
3.45 (0.08 to 12.83) |
1016.95 (21.67 to 3815.86) |
1.80 (0.04 to 6.80) |
8.89 (0.21 to 33.01) |
-2.01 (-2.25 to -1.77) |
Jilin |
468.81 (10.31 to 1808.05) |
3.02 (0.07 to 11.66) |
530.16 (11.84 to 1939.54) |
1.25 (0.03 to 4.65) |
8.75 (0.21 to 32.75) |
-3.19 (-3.62 to -2.75) |
Liaoning |
859.80 (19.15 to 3286.72) |
3.08 (0.07 to 11.86) |
1355.75 (29.84 to 5145.03) |
1.76 (0.04 to 6.73) |
8.59 (0.21 to 32.39) |
-1.55 (-1.8 to -1.31) |
Macao |
4.15 (0.09 to 15.17) |
1.52 (0.03 to 5.52) |
5.83 (0.12 to 22.49) |
0.63 (0.01 to 2.42) |
8.83 (0.21 to 32.97) |
-3.14 (-3.34 to -2.95) |
Ningxia |
113.15 (2.47 to 438.24) |
4.82 (0.11 to 18.84) |
243.82 (5.48 to 937.69) |
3.41 (0.08 to 13.20) |
8.44 (0.21 to 32.27) |
-0.98 (-1.13 to -0.82) |
Qinghai |
127.81 (2.69 to 476.30) |
5.44 (0.12 to 20.04) |
273.51 (5.83 to 1019.98) |
4.36 (0.10 to 16.29) |
8.88 (0.21 to 32.97) |
-0.51 (-0.71 to -0.3) |
Shaanxi |
796.28 (17.31 to 2954.11) |
3.61 (0.08 to 13.51) |
1044.35 (22.34 to 3865.56) |
1.98 (0.04 to 7.39) |
8.84 (0.21 to 32.83) |
-1.94 (-2.31 to -1.58) |
Shandong |
2363.43 (51.86 to 8795.77) |
3.73 (0.08 to 14.01) |
3543.69 (78.12 to 13024.72) |
2.31 (0.05 to 8.52) |
8.86 (0.21 to 32.97) |
-1.35 (-1.88 to -0.82) |
Shanghai |
492.02 (10.58 to 1852.30) |
3.50 (0.08 to 13.27) |
566.30 (12.04 to 2139.24) |
1.34 (0.03 to 5.06) |
8.70 (0.21 to 32.66) |
-3.34 (-3.43 to -3.25) |
Shanxi |
926.59 (20.74 to 3533.39) |
4.64 (0.11 to 17.87) |
1494.43 (32.86 to 5639.88) |
3.16 (0.07 to 12.10) |
8.59 (0.21 to 32.38) |
-1.05 (-1.52 to -0.57) |
Sichuan |
2491.59 (54.53 to 9295.86) |
3.11 (0.07 to 11.65) |
2747.61 (59.9 to 10112.62) |
2.09 (0.05 to 7.76) |
8.70 (0.21 to 32.57) |
-1.14(-1.91 to -0.36) |
Tianjin |
144.33 (3.25 to 539.00) |
2.04 (0.05 to 7.69) |
199.11 (4.28 to 773.61) |
0.94 (0.02 to 3.63) |
8.60 (0.21 to 32.48) |
-2.49 (-2.67 to -2.32) |
Tibet |
60.52 (1.32 to 228.81) |
4.12 (0.09 to 15.49) |
62.28 (1.31 to 227.56) |
2.35 (0.05 to 8.60) |
8.92 (0.21 to 32.95) |
-2.13 (-2.37 to -1.89) |
Xinjiang |
269.27 (5.72 to 1019.43) |
3.19 (0.07 to 12.12) |
559.31 (11.97 to 2077.37) |
2.34 (0.05 to 8.76) |
8.88 (0.21 to 32.84) |
-0.75 (-0.96 to -0.54) |
Yunnan |
385.05 (8.44 to 1452.29) |
1.59 (0.04 to 6.00) |
677.78 (14.8 to 2498.95) |
1.24 (0.03 to 4.61) |
8.71 (0.21 to 32.70) |
-0.36 (-0.69 to -0.02) |
Zhejiang |
1255.84 (27 to 4654.13) |
3.68 (0.08 to 13.67) |
1634.26 (34.75 to 5953.92) |
1.85 (0.04 to 6.76) |
9.00 (0.21 to 33.13) |
-2.14 (-2.44 to -1.83) |
DALYs=disability-adjusted life years; ASDR=age-standardized DALYs rate; PAF=population attributable fraction; EAPC=estimated annual percentage change.

Reviewer 4 Report
Comments and Suggestions for Authors
It is an epidemiological study on the role of high sodium intake in gastric cancer develoment, in different regions in China, through the time period 1990-2019
Interesting work, but reading and understanding the text turns out to be a difficult mental exercise: to many abbreviations and epidemiological indexes and statistical parameters!
As a reader, not familiar to such studies [I am a surgeon], I would like to have a table or box, explaining the main epidemiological/statistical terms commonly used.
The Tables, as presented are difficult in understanding; perhaps by using the landscape orientation it will become more easier readable - not each parameter to cover 2 or 3 lines. You have covered 7 pages with tables where you anayze gender [2 parameters] socio-demographic index [4 parameters] and regions [33 parameters]. I suggest to present socio-demographic indexes as an histogram. Regarding the 33 regions of China [I understand it is a huge area] it make to me no sence that Fujian and Hubei have similar nomber of deaths [1042 and 1035]; I suggest to classify regions according to the socio-demographic indexes and present the results more conclusivelly and more unestandable.
Regarding figure 2, make each map larger - one map after the other per group of lined, to be larger and easier to read -use more briliant colors and avoid the rectagular presenting the place of china in the world map
Finally, I need a paragraph given more information of the sourses of intaking salt in relation to the region and the dietary habits of inhabitants; do all eat preserved foods equally? whats the difference in eating preferences [or needs] between Tibet and Mongolia [to say some] on the one hand and Hong Kong and other sea-front areas? It there a relationship between geographic area/temperature and salt intake? May you present the two extremely differerent regions in gastric cancer frequency and analyze what makes the differnce? Generally speaking, I need to read something more understandable, besides the high epidemiological data you presents; dont forget, you try to publish in a Nutrition journal
Comments on the Quality of English Language
there are some misspellings
Author Response
For research article
Response to Reviewer X Comments
|
||
1. Summary |
|
|
Thank you very much for taking the time to review this manuscript. Please find the detailed responses below and the corresponding revisions/corrections highlighted/in track changes in the re-submitted files.
|
||
2. Questions for General Evaluation |
Reviewer’s Evaluation |
Response and Revisions |
Does the introduction provide sufficient background and include all relevant references? |
Can be improved |
Thank you for pointing this out. We have made changes. |
Are all the cited references relevant to the research? |
Yes |
|
Is the research design appropriate? |
Yes |
|
Are the methods adequately described? |
Can be improved |
Thank you for pointing this out. We have made some changes. |
Are the results clearly presented? |
Must be improved |
Thank you for pointing this out. We have made critical changes. |
Are the conclusions supported by the results? |
Yes |
|
3. Point-by-point response to Comments and Suggestions for Authors |
||
Comments 1: The Tables, as presented are difficult in understanding; perhaps by using the landscape orientation it will become more easier readable - not each parameter to cover 2 or 3 lines. You have covered 7 pages with tables where you anayze gender [2 parameters] socio-demographic index [4 parameters] and regions [33 parameters]. I suggest to present socio-demographic indexes as an histogram. Regarding the 33 regions of China [I understand it is a huge area] it make to me no sence that Fujian and Hubei have similar nomber of deaths [1042 and 1035]; I suggest to classify regions according to the socio-demographic indexes and present the results more conclusivelly and more unestandable. |
||
Response 1:Thank you for pointing this out. We agree with this comment. As you suggested, we have arranged the tables in order and splitted the original Table1 into Table1 and Supplementary Tables1, and we also splitted the original Table2 into Table2 and Supplementary Table2. The information of Supplementary Table1 and Supplementary Table2 could be available via certain website page. |
||
Comments 2: Regarding figure 2, make each map larger - one map after the other per group of lined, to be larger and easier to read -use more briliant colors and avoid the rectagular presenting the place of china in the world map |
||
Response 2: Agree. We have, accordingly, changed the order and size of the map to emphasize this point. – page 3. |
||
Comments 3: Finally, I need a paragraph given more information of the sourses of intaking salt in relation to the region and the dietary habits of inhabitants; do all eat preserved foods equally? whats the difference in eating preferences [or needs] between Tibet and Mongolia [to say some] on the one hand and Hong Kong and other sea-front areas? It there a relationship between geographic area/temperature and salt intake? May you present the two extremely differerent regions in gastric cancer frequency and analyze what makes the differnce? Generally speaking, I need to read something more understandable, besides the high epidemiological data you presents; dont forget, you try to publish in a Nutrition journal |
||
Response 3: Thank you for pointing this out. We agree with this comment. As you suggested, we have discussed the differences of dietary habits in different regions. The contents are as follows: ”Our results showed variations in the age-standardized mortality and DALY rates from 1990 to 2019 at the regional levels. There were considerable variations in the attributable burden of high sodium intake across regions. The disease burden in Qinghai and Shandong in 2019 were relatively the highest. This could be taken into account low SES there, a characteristic concerning a wide part of population. In particular, in low-income and middle-income regions sodium intake is more diffuse, diverse, and higher compared to those high-income regions[28]. The local residents in Shandong, one region close to The Yellow River Delta, are prone to have sea-foods and pickled foods with more sodium ingredients. Additionally, eating out has become popular than ever before and restaurant dishes often contain high salt. Also, the consumption of prepackaged food continues to increase and increases the exposure to high-sodium[32]. Whereas, the lowest disease burden were detected in Hong Kong and Macao across China. This may be related to the light diet for residents in Hong Kong and Macao. Those population consume less processed meat and pickled foods since the residents pay more attention to dietary patterns. Lifestyle modifications, alcohol drinking and high fat diet, could boost the effects of high sodium intake on gastric cancer and exert mediating effects. Diminishing the sodium intake would also scale back gastric cancer risk due to the emerging interaction effect. Dietary preference in daily life and demographic variables, including expanded life expectancy and aging, might be involved in serious disease burden of gastric cancer. ”– page 9, line 406 to 425. |
||
4. Response to Comments on the Quality of English Language |
||
Point 1:there are some misspellings |
||
Response 1: Thanks for your suggestions. We have tried our best to polish the language in the revised manuscript. |
||
5. Additional clarifications |
||
|
Supplementary Tables 1. Deaths and ASMR of gastric cancer disease attributable-high sodium intake in 1990 and 2019 and the temporal trends from 1990-2019.
Characteristics |
1990 |
2019 |
1990-2019 |
|||
|
Deaths cases, No. (95% UI) |
ASMR per 100,000 No.(95% UI) |
Deaths cases, No. (95% UI) |
ASMR per 100,000 No. (95% UI) |
PAFs % (95% UI) |
EAPC(%) in ASMR No. (95% CI) |
Region |
||||||
Anhui |
2146.88 (46.72 to 7905.63) |
5.49 (0.12 to 20.37) |
2620.63 (55.77 to 9898) |
2.87 (0.06 to 10.85) |
8.89 (0.21 to 33.04) |
-2.03 (-2.44 to -1.63) |
Beijing |
174.73 (3.86 to 651.83) |
1.98 (0.04 to 7.31) |
252.11 (5.41 to 921.33) |
0.78 (0.02 to 2.85) |
8.90 (0.21 to 32.91) |
-3.32 (-3.41 to -3.23) |
Chongqing |
278.69 (6.04 to 1032.87) |
2.36 (0.05 to 8.71) |
643.60 (13.83 to 2461.93) |
1.52 (0.03 to 5.85) |
8.69 (0.21 to 32.50) |
-1.14 (-1.54 to -0.74) |
Fujian |
1042.99 (22.75 to 3940.85) |
5.22 (0.12 to 19.57) |
1234.75 (26.77 to 4643.99) |
2.47 (0.05 to 9.31) |
8.81(0.21 to 32.80) |
-2.63 (-2.72 to -2.54) |
Gansu |
612.58 (14.23 to 2391.08) |
4.46 (0.11 to 17.82) |
975.81 (22.56 to 3759.51) |
2.84 (0.07 to 10.99) |
8.04 (0.21 to 31.41) |
-1.44 (-1.89 to -0.98) |
Guangdong |
890.53 (19.77 to 3437.91) |
1.98 (0.04 to 7.65) |
1215.51 (27.48 to 4638.79) |
0.93 (0.02 to 3.55) |
8.59 (0.21 to 32.40) |
-2.23 (-2.68 to -1.77) |
Guangxi |
621.23 (13.68 to 2397.13) |
2.17 (0.05 to 8.51) |
1003.72 (22.01 to 3781.63) |
1.63 (0.04 to 6.17) |
8.46 (0.21 to 32.10) |
-0.38 (-0.83 to 0.07) |
Guizhou |
413.55 (9.05 to 1525.41) |
2.03 (0.05 to 7.56) |
672.22 (14.67 to 2516.46) |
1.57 (0.03 to 5.96) |
8.68 (0.21 to 32.59) |
-0.38 (-0.71 to -0.05) |
Hainan |
160.62 (3.58 to 615.73) |
3.60 (0.08 to 13.95) |
256.92 (5.75 to 989.52) |
2.30 (0.05 to 8.89) |
8.42 (0.21 to 32.14) |
-1.12 (-1.42 to -0.82) |
Hebei |
1509.07 (33.34 to 5589.76) |
3.41 (0.08 to 12.69) |
2080.59 (44.65 to 7700.39) |
2.04 (0.04 to 7.56) |
8.83 (0.21 to 32.91) |
-1.63 (-2.09 to -1.18) |
Heilongjiang |
589.98 (13.10 to 2252.24) |
2.98 (0.07 to 11.43) |
929.05 (20.52 to 3481.43) |
1.57 (0.04 to 6.02) |
8.52 (0.21 to 32.26) |
-1.76 (-2.05 to -1.46) |
Henan |
2611.86 (57.8 to 9561.77) |
4.28 (0.10 to 15.83) |
2950.71 (64.73 to 10986.83) |
2.41 (0.05 to 9.02) |
8.79 (0.21 to 32.91) |
-1.91 (-2.30 to -1.52) |
Hong Kong |
81.96 (1.92 to 299.89) |
1.49 (0.03 to 5.46) |
107.24 (2.29 to 409.98) |
0.72 (0.02 to 2.72) |
8.79 (0.21 to 32.88) |
-2.46 (-2.69 to -2.22) |
Hubei |
1035.70 (23.33 to 3810.60) |
2.77 (0.06 to 10.18) |
1492.15 (32.15 to 5669.98) |
1.73 (0.04 to 6.54) |
8.72 (0.21 to 32.71) |
-1.06 (-1.44 to -0.68) |
Hunan |
762.78 (17.22 to 2820.83) |
1.75 (0.04 to 6.47) |
1132.43 (24.55 to 4254.50) |
1.16 (0.03 to 4.36) |
8.75 (0.21 to 32.69) |
-0.96 (-1.25 to -0.66) |
Inner Mongolia |
357.90 (7.78 to 1336.06) |
2.86 (0.06 to 10.7) |
574.57 (12.24 to 2104.58) |
1.69 (0.04 to 6.19) |
8.88 (0.21 to 32.98) |
-1.73 (-2.08 to -1.38) |
Jiangsu |
2330.12 (52.07 to 8597.48) |
4.34 (0.10 to 16.1) |
3034.31 (65.72 to 11195.55) |
2.22 (0.05 to 8.17) |
8.82 (0.21 to 32.89) |
-2.14 (-2.65 to -1.62) |
Jiangxi |
846.71 (18.73 to 3142.73) |
3.45 (0.08 to 12.83) |
1016.95 (21.67 to 3815.86) |
1.80 (0.04 to 6.80) |
8.89 (0.21 to 33.01) |
-2.01 (-2.25 to -1.77) |
Jilin |
468.81 (10.31 to 1808.05) |
3.02 (0.07 to 11.66) |
530.16 (11.84 to 1939.54) |
1.25 (0.03 to 4.65) |
8.75 (0.21 to 32.75) |
-3.19 (-3.62 to -2.75) |
Liaoning |
859.80 (19.15 to 3286.72) |
3.08 (0.07 to 11.86) |
1355.75 (29.84 to 5145.03) |
1.76 (0.04 to 6.73) |
8.59 (0.21 to 32.39) |
-1.55 (-1.8 to -1.31) |
Macao |
4.15 (0.09 to 15.17) |
1.52 (0.03 to 5.52) |
5.83 (0.12 to 22.49) |
0.63 (0.01 to 2.42) |
8.83 (0.21 to 32.97) |
-3.14 (-3.34 to -2.95) |
Ningxia |
113.15 (2.47 to 438.24) |
4.82 (0.11 to 18.84) |
243.82 (5.48 to 937.69) |
3.41 (0.08 to 13.20) |
8.44 (0.21 to 32.27) |
-0.98 (-1.13 to -0.82) |
Qinghai |
127.81 (2.69 to 476.30) |
5.44 (0.12 to 20.04) |
273.51 (5.83 to 1019.98) |
4.36 (0.10 to 16.29) |
8.88 (0.21 to 32.97) |
-0.51 (-0.71 to -0.3) |
Shaanxi |
796.28 (17.31 to 2954.11) |
3.61 (0.08 to 13.51) |
1044.35 (22.34 to 3865.56) |
1.98 (0.04 to 7.39) |
8.84 (0.21 to 32.83) |
-1.94 (-2.31 to -1.58) |
Shandong |
2363.43 (51.86 to 8795.77) |
3.73 (0.08 to 14.01) |
3543.69 (78.12 to 13024.72) |
2.31 (0.05 to 8.52) |
8.86 (0.21 to 32.97) |
-1.35 (-1.88 to -0.82) |
Shanghai |
492.02 (10.58 to 1852.30) |
3.50 (0.08 to 13.27) |
566.30 (12.04 to 2139.24) |
1.34 (0.03 to 5.06) |
8.70 (0.21 to 32.66) |
-3.34 (-3.43 to -3.25) |
Shanxi |
926.59 (20.74 to 3533.39) |
4.64 (0.11 to 17.87) |
1494.43 (32.86 to 5639.88) |
3.16 (0.07 to 12.10) |
8.59 (0.21 to 32.38) |
-1.05 (-1.52 to -0.57) |
Sichuan |
2491.59 (54.53 to 9295.86) |
3.11 (0.07 to 11.65) |
2747.61 (59.9 to 10112.62) |
2.09 (0.05 to 7.76) |
8.70 (0.21 to 32.57) |
-1.14(-1.91 to -0.36) |
Tianjin |
144.33 (3.25 to 539.00) |
2.04 (0.05 to 7.69) |
199.11 (4.28 to 773.61) |
0.94 (0.02 to 3.63) |
8.60 (0.21 to 32.48) |
-2.49 (-2.67 to -2.32) |
Tibet |
60.52 (1.32 to 228.81) |
4.12 (0.09 to 15.49) |
62.28 (1.31 to 227.56) |
2.35 (0.05 to 8.60) |
8.92 (0.21 to 32.95) |
-2.13 (-2.37 to -1.89) |
Xinjiang |
269.27 (5.72 to 1019.43) |
3.19 (0.07 to 12.12) |
559.31 (11.97 to 2077.37) |
2.34 (0.05 to 8.76) |
8.88 (0.21 to 32.84) |
-0.75 (-0.96 to -0.54) |
Yunnan |
385.05 (8.44 to 1452.29) |
1.59 (0.04 to 6.00) |
677.78 (14.8 to 2498.95) |
1.24 (0.03 to 4.61) |
8.71 (0.21 to 32.70) |
-0.36 (-0.69 to -0.02) |
Zhejiang |
1255.84 (27 to 4654.13) |
3.68 (0.08 to 13.67) |
1634.26 (34.75 to 5953.92) |
1.85 (0.04 to 6.76) |
9.00 (0.21 to 33.13) |
-2.14 (-2.44 to -1.83) |
ASMR=age-standardized mortality rate; PAF=population attributable fraction; EAPC=estimated annual percentage change.
Supplementary Tables 2. DALYs and ASDR of gastric cancer disease attributable to high sodium intake in 1990 and 2019 and the temporal trends from 1990 to 2019.
Characteristics |
1990 |
2019 |
1990-2019 |
|||
|
Deaths cases, No. (95% UI) |
ASDR per 100,000 No.(95% UI) |
DALYs, No. (95% UI) |
ASDR per 100,000 No. (95% UI) |
PAFs % (95% UI) |
EAPC(%) in ASMR No. (95% CI) |
Region |
||||||
Anhui |
2146.88 (46.72 to 7905.63) |
5.49 (0.12 to 20.37) |
2620.63 (55.77 to 9898) |
2.87 (0.06 to 10.85) |
8.89 (0.21 to 33.04) |
-2.03 (-2.44 to -1.63) |
Beijing |
174.73 (3.86 to 651.83) |
1.98 (0.04 to 7.31) |
252.11 (5.41 to 921.33) |
0.78 (0.02 to 2.85) |
8.90 (0.21 to 32.91) |
-3.32 (-3.41 to -3.23) |
Chongqing |
278.69 (6.04 to 1032.87) |
2.36 (0.05 to 8.71) |
643.60 (13.83 to 2461.93) |
1.52 (0.03 to 5.85) |
8.69 (0.21 to 32.50) |
-1.14 (-1.54 to -0.74) |
Fujian |
1042.99 (22.75 to 3940.85) |
5.22 (0.12 to 19.57) |
1234.75 (26.77 to 4643.99) |
2.47 (0.05 to 9.31) |
8.81(0.21 to 32.80) |
-2.63 (-2.72 to -2.54) |
Gansu |
612.58 (14.23 to 2391.08) |
4.46 (0.11 to 17.82) |
975.81 (22.56 to 3759.51) |
2.84 (0.07 to 10.99) |
8.04 (0.21 to 31.41) |
-1.44 (-1.89 to -0.98) |
Guangdong |
890.53 (19.77 to 3437.91) |
1.98 (0.04 to 7.65) |
1215.51 (27.48 to 4638.79) |
0.93 (0.02 to 3.55) |
8.59 (0.21 to 32.40) |
-2.23 (-2.68 to -1.77) |
Guangxi |
621.23 (13.68 to 2397.13) |
2.17 (0.05 to 8.51) |
1003.72 (22.01 to 3781.63) |
1.63 (0.04 to 6.17) |
8.46 (0.21 to 32.10) |
-0.38 (-0.83 to 0.07) |
Guizhou |
413.55 (9.05 to 1525.41) |
2.03 (0.05 to 7.56) |
672.22 (14.67 to 2516.46) |
1.57 (0.03 to 5.96) |
8.68 (0.21 to 32.59) |
-0.38 (-0.71 to -0.05) |
Hainan |
160.62 (3.58 to 615.73) |
3.60 (0.08 to 13.95) |
256.92 (5.75 to 989.52) |
2.30 (0.05 to 8.89) |
8.42 (0.21 to 32.14) |
-1.12 (-1.42 to -0.82) |
Hebei |
1509.07 (33.34 to 5589.76) |
3.41 (0.08 to 12.69) |
2080.59 (44.65 to 7700.39) |
2.04 (0.04 to 7.56) |
8.83 (0.21 to 32.91) |
-1.63 (-2.09 to -1.18) |
Heilongjiang |
589.98 (13.10 to 2252.24) |
2.98 (0.07 to 11.43) |
929.05 (20.52 to 3481.43) |
1.57 (0.04 to 6.02) |
8.52 (0.21 to 32.26) |
-1.76 (-2.05 to -1.46) |
Henan |
2611.86 (57.8 to 9561.77) |
4.28 (0.10 to 15.83) |
2950.71 (64.73 to 10986.83) |
2.41 (0.05 to 9.02) |
8.79 (0.21 to 32.91) |
-1.91 (-2.30 to -1.52) |
Hong Kong |
81.96 (1.92 to 299.89) |
1.49 (0.03 to 5.46) |
107.24 (2.29 to 409.98) |
0.72 (0.02 to 2.72) |
8.79 (0.21 to 32.88) |
-2.46 (-2.69 to -2.22) |
Hubei |
1035.70 (23.33 to 3810.60) |
2.77 (0.06 to 10.18) |
1492.15 (32.15 to 5669.98) |
1.73 (0.04 to 6.54) |
8.72 (0.21 to 32.71) |
-1.06 (-1.44 to -0.68) |
Hunan |
762.78 (17.22 to 2820.83) |
1.75 (0.04 to 6.47) |
1132.43 (24.55 to 4254.50) |
1.16 (0.03 to 4.36) |
8.75 (0.21 to 32.69) |
-0.96 (-1.25 to -0.66) |
Inner Mongolia |
357.90 (7.78 to 1336.06) |
2.86 (0.06 to 10.7) |
574.57 (12.24 to 2104.58) |
1.69 (0.04 to 6.19) |
8.88 (0.21 to 32.98) |
-1.73 (-2.08 to -1.38) |
Jiangsu |
2330.12 (52.07 to 8597.48) |
4.34 (0.10 to 16.1) |
3034.31 (65.72 to 11195.55) |
2.22 (0.05 to 8.17) |
8.82 (0.21 to 32.89) |
-2.14 (-2.65 to -1.62) |
Jiangxi |
846.71 (18.73 to 3142.73) |
3.45 (0.08 to 12.83) |
1016.95 (21.67 to 3815.86) |
1.80 (0.04 to 6.80) |
8.89 (0.21 to 33.01) |
-2.01 (-2.25 to -1.77) |
Jilin |
468.81 (10.31 to 1808.05) |
3.02 (0.07 to 11.66) |
530.16 (11.84 to 1939.54) |
1.25 (0.03 to 4.65) |
8.75 (0.21 to 32.75) |
-3.19 (-3.62 to -2.75) |
Liaoning |
859.80 (19.15 to 3286.72) |
3.08 (0.07 to 11.86) |
1355.75 (29.84 to 5145.03) |
1.76 (0.04 to 6.73) |
8.59 (0.21 to 32.39) |
-1.55 (-1.8 to -1.31) |
Macao |
4.15 (0.09 to 15.17) |
1.52 (0.03 to 5.52) |
5.83 (0.12 to 22.49) |
0.63 (0.01 to 2.42) |
8.83 (0.21 to 32.97) |
-3.14 (-3.34 to -2.95) |
Ningxia |
113.15 (2.47 to 438.24) |
4.82 (0.11 to 18.84) |
243.82 (5.48 to 937.69) |
3.41 (0.08 to 13.20) |
8.44 (0.21 to 32.27) |
-0.98 (-1.13 to -0.82) |
Qinghai |
127.81 (2.69 to 476.30) |
5.44 (0.12 to 20.04) |
273.51 (5.83 to 1019.98) |
4.36 (0.10 to 16.29) |
8.88 (0.21 to 32.97) |
-0.51 (-0.71 to -0.3) |
Shaanxi |
796.28 (17.31 to 2954.11) |
3.61 (0.08 to 13.51) |
1044.35 (22.34 to 3865.56) |
1.98 (0.04 to 7.39) |
8.84 (0.21 to 32.83) |
-1.94 (-2.31 to -1.58) |
Shandong |
2363.43 (51.86 to 8795.77) |
3.73 (0.08 to 14.01) |
3543.69 (78.12 to 13024.72) |
2.31 (0.05 to 8.52) |
8.86 (0.21 to 32.97) |
-1.35 (-1.88 to -0.82) |
Shanghai |
492.02 (10.58 to 1852.30) |
3.50 (0.08 to 13.27) |
566.30 (12.04 to 2139.24) |
1.34 (0.03 to 5.06) |
8.70 (0.21 to 32.66) |
-3.34 (-3.43 to -3.25) |
Shanxi |
926.59 (20.74 to 3533.39) |
4.64 (0.11 to 17.87) |
1494.43 (32.86 to 5639.88) |
3.16 (0.07 to 12.10) |
8.59 (0.21 to 32.38) |
-1.05 (-1.52 to -0.57) |
Sichuan |
2491.59 (54.53 to 9295.86) |
3.11 (0.07 to 11.65) |
2747.61 (59.9 to 10112.62) |
2.09 (0.05 to 7.76) |
8.70 (0.21 to 32.57) |
-1.14(-1.91 to -0.36) |
Tianjin |
144.33 (3.25 to 539.00) |
2.04 (0.05 to 7.69) |
199.11 (4.28 to 773.61) |
0.94 (0.02 to 3.63) |
8.60 (0.21 to 32.48) |
-2.49 (-2.67 to -2.32) |
Tibet |
60.52 (1.32 to 228.81) |
4.12 (0.09 to 15.49) |
62.28 (1.31 to 227.56) |
2.35 (0.05 to 8.60) |
8.92 (0.21 to 32.95) |
-2.13 (-2.37 to -1.89) |
Xinjiang |
269.27 (5.72 to 1019.43) |
3.19 (0.07 to 12.12) |
559.31 (11.97 to 2077.37) |
2.34 (0.05 to 8.76) |
8.88 (0.21 to 32.84) |
-0.75 (-0.96 to -0.54) |
Yunnan |
385.05 (8.44 to 1452.29) |
1.59 (0.04 to 6.00) |
677.78 (14.8 to 2498.95) |
1.24 (0.03 to 4.61) |
8.71 (0.21 to 32.70) |
-0.36 (-0.69 to -0.02) |
Zhejiang |
1255.84 (27 to 4654.13) |
3.68 (0.08 to 13.67) |
1634.26 (34.75 to 5953.92) |
1.85 (0.04 to 6.76) |
9.00 (0.21 to 33.13) |
-2.14 (-2.44 to -1.83) |
DALYs=disability-adjusted life years; ASDR=age-standardized DALYs rate; PAF=population attributable fraction; EAPC=estimated annual percentage change.

Round 2
Reviewer 1 Report
Comments and Suggestions for Authors
The suggestions made for the previous version have been addressed. The manuscript can be accepted.